# Proteostasis modulates gene dosage evolution in antibiotic-resistant bacteria

Chinmaya Jena, Saillesh Chinnaraj, Soham Deolankar, Nishad Matange*

Department of Biology, Indian Institute of Science Education and Research, Pune, India

## eLife Assessment

This **important** study explores the interplay between gene dosage and gene mutations in the evolution of antibiotic resistance. The authors provide **compelling** evidence connecting proteostasis with gene duplication during experimental evolution in a model system. This paper is likely to be of interest to researchers studying antibiotic resistance, proteostasis, and bacterial evolution.

**Abstract** Evolution of gene expression frequently drives antibiotic resistance in bacteria. We had previously (Patel and Matange, *eLife*, 2021) shown that, in *Escherichia coli*, mutations at the *mgrB* locus were beneficial under trimethoprim exposure and led to overexpression of dihydrofolate reductase (DHFR), encoded by the *folA* gene. Here, we show that DHFR levels are further enhanced by spontaneous duplication of a genomic segment encompassing *folA* and spanning hundreds of kilobases. This duplication was rare in wild-type *E. coli*. However, its frequency was elevated in a *lon*-knockout strain, altering the mutational landscape early during trimethoprim adaptation. We then exploit this system to investigate the relationship between trimethoprim pressure and *folA* copy number. During long-term evolution, *folA* duplications were frequently reversed. Reversal was slower under antibiotic pressure, first requiring the acquisition of point mutations in DHFR or its promoter. Unexpectedly, despite resistance-conferring point mutations, some populations under high trimethoprim pressure maintained *folA* duplication to compensate for low abundance DHFR mutants. We find that evolution of gene dosage depends on expression demand, which is generated by antibiotic and exacerbated by proteolysis of drug-resistant mutants of DHFR. We propose a novel role for proteostasis as a determinant of copy number evolution in antibiotic-resistant bacteria.

## Introduction

The dosage of a gene, i.e., relative copy number, directly impacts its level of expression and can alter organismal phenotype and fitness (*Andersson and Hughes, 2009*; *Elliott et al., 2013*; *Kondrashov, 2012*). Mutations that increase copy numbers, termed as gene duplications and amplifications (GDAs), are implicated in diverse biological phenomena including cancer progression (*Dürrbaum and Storchová, 2016*; *Jiang et al., 2013*; *Spielmann et al., 2018*), speciation (*Pascual-Anaya et al., 2013*; *Taylor et al., 2001*; *Ting et al., 2004*), reproductive isolation (*Jiao et al., 2021*; *Maclean and Greig, 2011*; *Zuellig and Sweigart, 2018*), and antimicrobial resistance (*Elliott et al., 2013*; *Sandegren and Andersson, 2009*). GDAs are important for rapid adaptation since their frequencies of occurrence can be higher by several orders of magnitude than point mutations (*Elliott et al., 2013*; *Sandegren and Andersson, 2009*; *Anderson and Roth, 1981*; *Seaton et al., 2012*; *Sonti and Roth, 1989*). Following the initial GDA mutation, however, amplified genes could be lost over time, pseudogenised, sub-functionalised, or neo-functionalised (*Hahn, 2009*; *Innan and Kondrashov, 2010*; *Kuzmin et al., 2022*). Which of these possibilities is realised depends on the nature and strength of

*For correspondence:
nishad@iiserpune.ac.in

Competing interest: The authors declare that no competing interests exist.

selection imposed by the environment, as well as the fitness cost associated with maintaining multiple gene copies (*Hahn, 2009*; *Innan and Kondrashov, 2010*; *Kuzmin et al., 2022*).

In bacteria, GDA mutations are frequently encountered during adaptation to antibiotics (*Sandegren and Andersson, 2009*) or heavy metals (*Hufnagel et al., 2021*; *von Rozycki and Nies, 2009*). They occur by RecA-dependent nonequal recombination as well as RecA-independent processes and often involve repeat sequences such as in IS-elements (*Bi and Liu, 1994*; *Reams et al., 2012*; *Reams et al., 2010*; *Reams and Neidle, 2003*). Amplified genes in drug-resistant bacteria code for effectors of resistance such as drug inactivating enzymes, drug target modifiers, or 'bypass' mechanisms (*Min et al., 2008*; *Sun et al., 2009*; *Vinella et al., 2000*), but may also compensate for resistance-associated fitness costs (*Nilsson et al., 2006*). More recently, the view that GDAs are less stable than point mutations has emerged. For instance, transient alterations in gene dosage are a common mechanism of heteroresistance, i.e., resistant subpopulations within a drug-sensitive bacterial isolate (*Hufnagel et al., 2021*; *Andersson et al., 2019*; *Nicoloff et al., 2019*). Further, it is estimated that as many as 20% of cells in a bacterial population may harbour a duplication of some genomic region (*Sun et al., 2012*). These copy number polymorphisms create within-population heterogeneities in gene expression, serving as a bet-hedging strategy with adaptive value in fluctuating environments (*Tomanek et al., 2020*).

Dihydrofolate reductase (DHFR) is a ubiquitous enzyme crucial for growth and division. It catalyses the NADPH-dependent reduction of tetrahydrofolate, an important cofactor in nucleotide and amino acid biosynthesis. DHFR is coded by an essential gene in most organisms and its deletion or pharmacological inhibition results in cell death or growth stasis (*Ahmad et al., 1998*). Not surprisingly, antifolates are effective therapeutic agents for diverse pathologies (*Gangjee et al., 2007*; *Gangjee et al., 2008*). For instance, methotrexate, used in the treatment of cancers and autoimmune conditions (*Huennekens, 1994*), pyrimethamine, used as an antimalarial (*Plowe, 2022*), and trimethoprim, used as an antibacterial (*Gleckman et al., 1981*), are all competitive inhibitors of DHFR. Interestingly, copy number amplification and consequent overexpression of DHFR confers resistance in all three systems (*Brochet et al., 2008*; *Göker et al., 1995*; *Inselburg et al., 1987*; *Morales et al., 2009*). Thus, the gene dosage of DHFR appears to be tightly regulated across evolutionarily distant organisms, making it a useful system to understand the principles underlying selection and evolutionary fate of GDA mutations.

In *Escherichia coli* DHFR is coded by *folA*. Several point mutations that confer trimethoprim resistance have been identified in this enzyme (*Baym et al., 2016*; *Palmer et al., 2015*; *Patel and Matange, 2021*; *Watson et al., 2007*). Additionally, mechanisms that transcriptionally upregulate *folA* expression are also known in trimethoprim-resistant *E. coli*. These include *cis*-regulatory mutations in the *folA* promoter and *trans*-acting loss-of-function mutations in *mgrB*, which upregulate *folA* transcription in a PhoQP-dependent manner (*Baym et al., 2016*; *Palmer et al., 2015*; *Patel and Matange, 2021*; *Toprak et al., 2011*). Earlier work from our group identified Lon, a conserved quality control protease in bacteria, as a posttranslational regulator of trimethoprim resistance. Lon degraded mutant DHFRs and reduced their half-lives in vivo, directly affecting genotype-phenotype relationships of resistance-conferring mutations (*Matange, 2020*; *Rodrigues et al., 2016*).

In this study, we first compare early mutational events in wild-type and Lon-deficient *E. coli* challenged with trimethoprim. We show that a large genomic duplication encompassing *folA* confers trimethoprim resistance in both genetic backgrounds, though its frequency of occurrence is elevated in *lon*-knockout *E. coli*. We then use this system to ask what factors determine the fate of duplicated *folA* gene copies during evolution at different trimethoprim pressures.

## Results

### A large IS-flanked duplication encompassing *folA* mediates rapid adaptation to trimethoprim in *E. coli*

We had previously evolved five replicate populations of wild-type *E. coli* K-12 MG1655 in 300 ng/mL of trimethoprim (~0.3× MIC) for 25 generations. We then isolated and sequenced the genome of a dominant trimethoprim-resistant clone from each population (designated as WTMPR1–5) (*Figure 1A*). These sequencing results have been reported earlier (*Patel and Matange, 2021*). Briefly, all isolates harboured loss-of-function mutations in the *mgrB* gene (*Figure 1B*) that resulted in PhoQP-dependent

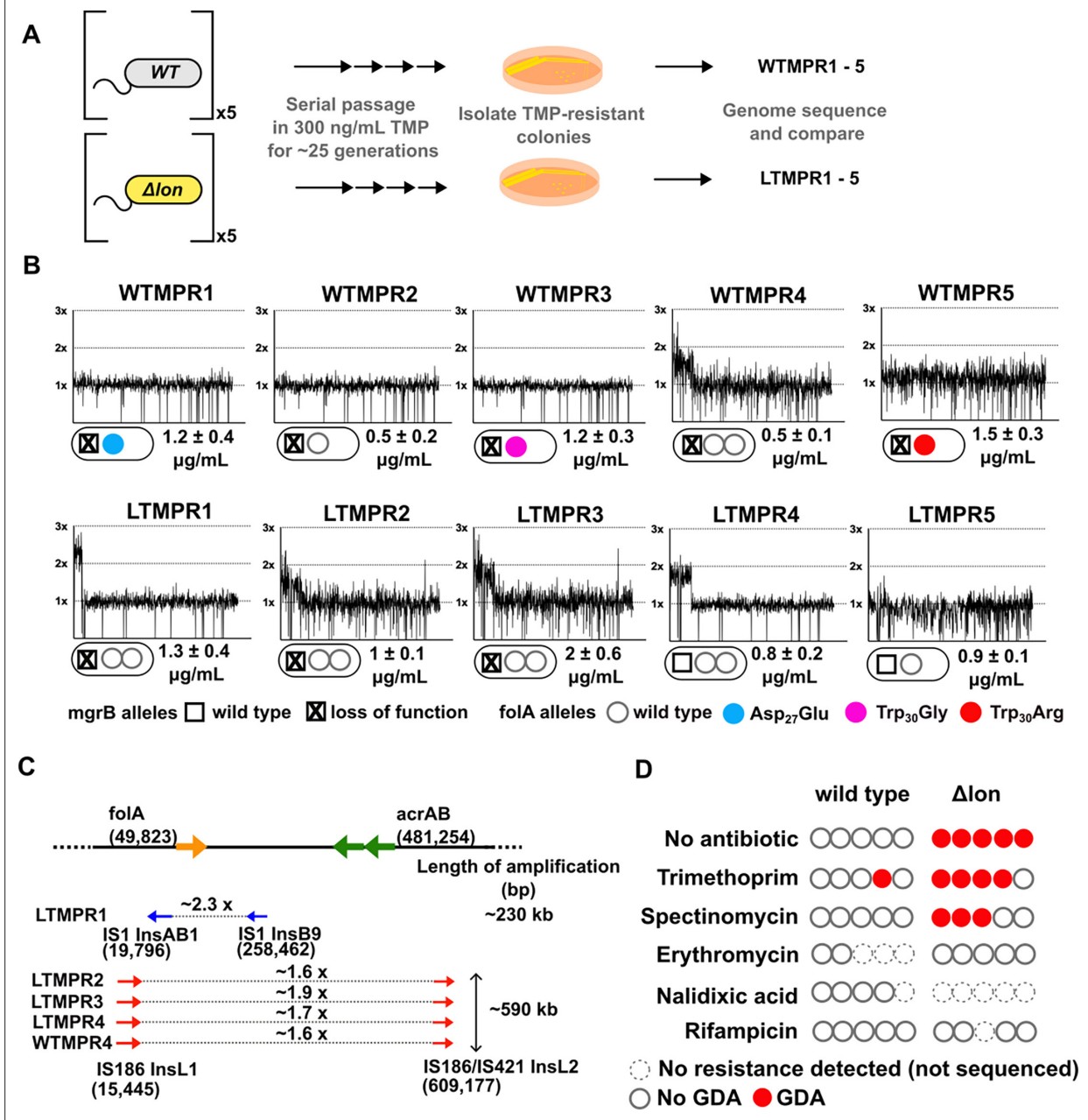

**Figure 1.** Adaptation to antibiotics by large genomic duplications is favoured in *lon*-deficient *E. coli*. (**A**) Schematic of the experimental pipeline used to compare the mutation repertoire of adaptation to trimethoprim in wild-type or *lon*-deficient *E. coli*. (**B**) Coverage depth plots and mutations at the *folA* and *mgrB* loci of five independently evolved trimethoprim-resistant isolates derived from wild-type (WTMPR1–5) or *Δlon E. coli* (LTMPR1–5). Coverage depth plots show the number of reads from Illumina short-read sequencing (y-axis) mapped to each genomic coordinate of the reference genome (x-axis). Coverage of 1×, 2×, and 3× are marked by dotted lines for reference. Mutations in *mgrB* and *folA* and trimethoprim inhibitory concentration 50 (IC50) values (mean ± SD from three independent measurements) are provided for each isolate below the appropriate coverage depth plot. (**C**) Cartoon (not to scale) showing the duplicated genomic stretch and flanking IS-elements in trimethoprim-resistant isolates. Blue arrows represent IS1, while red arrows represent IS186. The positions of *folA* and *acrAB* genes, implicated in trimethoprim resistance in earlier studies, relative to the duplicated stretch are shown. Fold increase in the number of reads corresponding to the *folA* gene are indicated for each isolate. (**D**) Summary of the prevalence of gene duplication and amplification (GDA) mutations detected in wild-type or *E. coli Δlon* after ~25 generations of evolution in control media (no antibiotic) or in media supplemented with sub-MIC antibiotics. Each circle represents one of the five replicates in the evolution experiment. Empty circles represent replicates in which no resistance was detected (dotted perimeter) or where the whole population or resistant isolates were sequenced but no GDA mutation was detected (solid perimeter). Red filled circles indicate that a GDA mutation was identified.

The online version of this article includes the following source data and figure supplement(s) for figure 1:

*Figure 1 continued on next page*

*Figure 1 continued*

**Figure supplement 1.** Loss of *mgrB* or *pitA* genes is beneficial in trimethoprim.

**Figure supplement 2.** Stability of heterologously expressed InsB and InsL transposases in wild-type or Δ*lon E. coli* measured using a chloramphenicol (CMP) chase assay.

**Figure supplement 2—source data 1.** Annotated image files for western blots in *Figure 1—figure supplement 2*.

**Figure supplement 2—source data 2.** Raw image file for western blots in *Figure 1—figure supplement 2*.

**Figure supplement 3.** Coverage plots and flanking IS-elements for *E. coli* Δ*lon* populations evolving in no antibiotic or spectinomycin that showed gene duplication and amplification (GDA) mutations.

overexpression of *folA* (**Patel and Matange, 2021**). Additionally, isolates WTMPR1, 3, and 5 had mutations at the active site of DHFR, while WTMPR4 harboured a large genomic duplication of ~0.6 Mb encompassing *folA* (**Figure 1B and C**). We performed similar evolution experiments starting with an isogenic *E. coli* Δ*lon* strain to ask if early adaptation to trimethoprim was altered by Lon deficiency (**Figure 1A**). Wild-type and *lon*-deficient *E. coli* have similar MIC for trimethoprim, though the *lon*-knockout has a higher inhibitory concentration 50 (IC50) (**Matange, 2020**). Among trimethoprim-resistant isolates derived from *E. coli* Δ*lon*, designated as LTMPR1–5, three harboured mutations at the *mgrB* locus (LTMPR1–3, **Figure 1A and B**, **Table 1**). LTMPR4 and LTMPR5 harboured mutations in *pitA*, coding for a metal-phosphate symporter, that have been reported earlier in trimethoprim adapted *E. coli* (**Vinchhi et al., 2023**). LTMPR5 had a deletion encompassing *pitA*, indicating that loss of PitA was beneficial in trimethoprim. A knockout of *pitA* indeed showed greater colony formation on trimethoprim-supplemented media demonstrating that mutations in this gene were adaptive (**Figure 1—figure supplement 1**), though the underlying molecular mechanisms are unknown at present. Interestingly, none of the LTMPR isolates had mutations in DHFR or its promoter. Instead, LTMPR2–4 had evolved GDA mutations similar to WTMPR4, while LTMPR1 had a duplication in an overlapping but shorter genomic stretch (**Figure 1B and C**, **Table 1**).

Duplicated genomic regions in WTMPR4 and LTMPR1–4 were flanked by IS-elements (**Figure 1C**). Seeking to explain the higher frequency of GDA mutations in the trimethoprim-adapted *lon*-knockout, we asked whether IS-transposases were sensitive to proteolysis by Lon. Indeed, heterologously expressed InsB and InsL transposases had greater stability in the *lon*-knockout (**Figure 1—figure supplement 2**), suggesting that their accumulation may contribute to higher GDA frequency in *E. coli* Δ*lon*. It would follow that a higher proportion of GDA mutants would be isolated from *E. coli* Δ*lon* during adaptation in other environments as well. In agreement, GDA mutations were detected after evolution of *E. coli* Δ*lon* in two of the five additional growth conditions tested (**Figure 1D**, **Figure 1—figure supplement 3**). In all cases duplications were flanked by IS-elements (**Figure 1—figure supplement 3**). Further, duplicated genomic regions were similar among isolates derived from the same growth condition, but different across environments (**Figure 1B**, **Figure 1—figure supplement 3**). On

**Table 1.** Complete list of mutations identified in isolates LTMPR1–5.

| Gene/locus | LTMPR1 | LTMPR2 | LTMPR3 | LTMPR4 | LTMPR5 | Description |
|---|---|---|---|---|---|---|
| *mgrB* | IS5 insertion at –46 | IS2 insertion at +82 | IS5 insertion at –36 | – | – | Negative feedback regulator of PhoQ sensor kinase |
| *ybcK* | – | IS2 insertion at +118 | – | – | – | Prophage recombinase |
| *pitA* | – | – | – | Ser$_{425}$Pro | Deletion of 18 genes [yhiN, pitA… yhiS]. IS5-mediated | Metal/phosphate symporter |
| *insH21* | Δ *insH21* | Δ *insH21* | Δ *insH21* | Δ *insH21* | Δ *insH21* | IS5 transposase |
| GDA (encompassing *folA*) | 19,796–258,462 bp (~2.3×) | 15,445–609,177 bp (~1.6×) | 15,445–609,177 bp (~1.9×) | 15,445–609,177 bp (~1.7×) | – | |

the other hand, wild-type *E. coli* did not show GDA mutations in any of the other tested conditions (*Figure 1D*).

The duplicated region in trimethoprim-resistant isolates included genes previously associated with resistance to trimethoprim, namely, *folA*, and *acrA* and *B*, which code for components of the AcrAB-TolC efflux pump (*Figure 1C*). In one of the isolates, i.e., LTMPR1, the amplified stretch encompassed *folA* but not *acrAB* without affecting trimethoprim IC50 values (*Figure 1B and C*). We therefore reasoned that duplication of *folA* was the main effector of adaptation to trimethoprim. Immunoblotting showed that DHFR protein was indeed overproduced in isolates LTMPR1 and WTMPR4 over and above the increase mediated by inactivation of *mgrB* alone (*Figure 2A*).

In addition to *folA*, ~173 and ~531 other genes were also duplicated in LTMPR1 and WTMPR4, respectively. To understand the global transcriptional burden imposed by this amplification, we analysed the gene expression profiles of WTMPR4 and LTMPR1 using RNA-sequencing (*Supplementary file 1*). Since both isolates had loss-of-function mutations in *mgrB*, which cause large changes in global gene expression (*Vinchhi et al., 2023*), we also analysed the transcriptomes of *E. coli ΔmgrB* and *E. coli ΔlonΔmgrB* strains for comparison (*Supplementary file 1*). Transcript level of *folA* was higher in LTMPR1 (2.3-fold) and WTMPR4 (5.7-fold) over respective controls (*Supplementary file 1*). Other duplicated genes were mildly overexpressed by ~1.7-fold on average in LTMPR1 (over *E. coli ΔlonΔmgrB*) and ~1.3-fold in WTMPR4 (over *E. coli ΔmgrB*) (*Figure 2B*, *Supplementary file 1*). However, there was large variation in expression level among duplicated genes (*Figure 2B*), indicating that increase in gene dosage did not necessarily result in enhanced expression levels. Instead, global gene expression profiles of WTMPR4 and LTMPR1 were correlated with *E. coli ΔmgrB* and *E. coli ΔlonΔmgrB,* respectively (*Figure 2C and D*) and most of the top 50 up/downregulated genes in WTMPR4 and LTMPR1 could be attributed to loss of *mgrB* alone (*Figure 2—figure supplement 1*). In line with this result, strains with and without genomic amplifications had comparable fitness costs (~20%) in drug-free media, similar in magnitude to the cost of *mgrB* inactivation (*Figure 2E*). Further, the fitness cost was alleviated in growth media supplemented with high Mg$^{2+}$, i.e., by inhibiting the PhoQP pathway, confirming the source of this cost to be loss of *mgrB* rather than the GDA (*Figure 2F*). Thus, we concluded that large GDA mutations encompassing *folA* led to DHFR overexpression and represented an alternate mechanism for trimethoprim resistance in *E. coli* with undetectable additional cost.

## Maintenance of *folA* duplication is linked to antibiotic-generated demand for DHFR expression

We next investigated the evolutionary fate of duplicate *folA* gene copies by evolving the LTMPR1 isolate, which harboured the shortest duplication, in the absence of antibiotic. We also evolved this isolate at sustained drug pressure, i.e., trimethoprim at the same concentration at which it had originally emerged (300 ng/mL). Using three replicate populations in each condition, we asked how DHFR expression and trimethoprim IC50 values changed over ~252 generations of evolution and used genome sequencing to identify the underlying mutational basis. As controls, we performed similar evolution experiments with WTMPR4 (which harboured a GDA and a functional Lon protease) and WTMPR5 (which harboured a *folA* point mutation and a functional Lon protease) (*Figure 3A*).

### Evolution in trimethoprim-free media

In drug-free conditions, trimethoprim IC50 of LTMPR1 rapidly declined and a concomitant loss of colony formation on trimethoprim-supplemented plates was observed (*Figure 3B and C*). Evolved WTMPR4, but not WTMPR5, showed similar loss of colony formation on trimethoprim, indicating that resistance mediated by GDA mutations was less stable than point mutations regardless of the presence of Lon (*Figure 3C*). For LTMPR1, these trends were explained by a decline in DHFR expression levels over evolution and reversion to a single *folA* copy (*Figure 3D, E*, *Figure 3—figure supplement 1*). Mutation in *mgrB* was, however, consistently maintained till the end of the experiment (*Supplementary file 2*). In agreement with genomics data, trimethoprim IC50s of evolving populations stabilised at a value similar to *E. coli ΔlonΔmgrB* (*Figure 3B*). Interestingly, a second GDA mutation ('GDA-2', ~2.2–3.3 Mb) emerged in these populations (*Figure 3E*, *Figure 3—figure supplement 1*). The GDA-2 mutation was apparently lost and regained multiple times during the experiment (*Figure 3E*, *Figure 3—figure supplement 1*), suggesting a dynamic maintenance in the population rather than

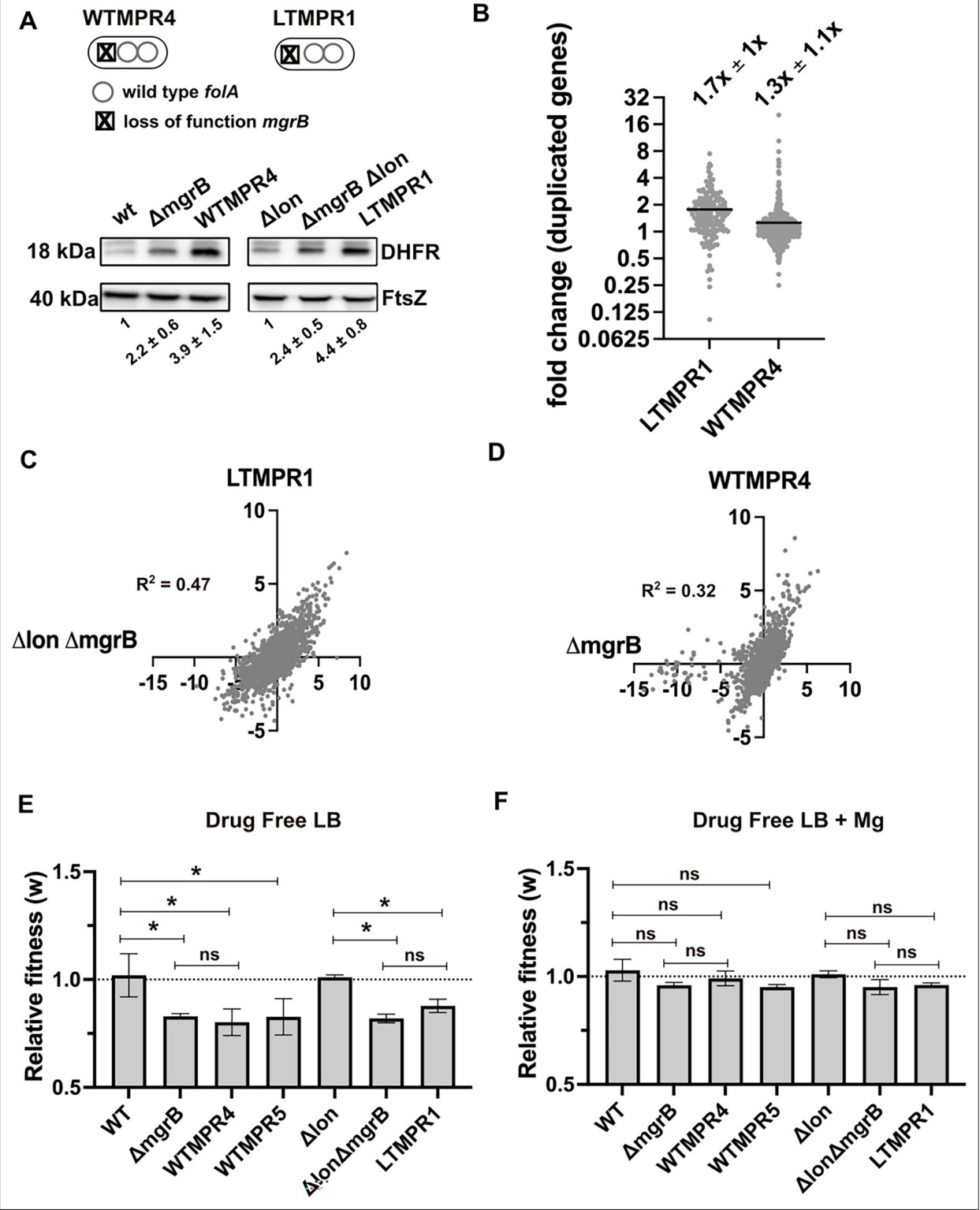

**Figure 2.** Overexpression of dihydrofolate reductase (DHFR) in trimethoprim-resistant isolates with genomic duplication. (**A**) DHFR protein expression in trimethoprim-resistant isolates WTMPR4 and LTMPR1, compared with respective ancestors and *mgrB*-knockout strains. DHFR protein was detected by immunoblotting using an anti-DHFR polyclonal antibody. FtsZ was used as loading control. A representative immunoblot from three biological replicates is shown. Quantitation was performed by calculating band intensities using image analysis. Fold change values of DHFR levels over controls (wild-type or *E. coli Δlon*, set to 1) are shown as mean ± SD from triplicate measurements. Alleles of *mgrB* and *folA* in WTMPR4 and LTMPR1 are shown diagrammatically. (**B**) Expression level of duplicated genes in LTMPR1 and WTMPR4, determined by RNA-sequencing, expressed as fold over *E. coli*

*Figure 2 continued on next page*

*Figure 2 continued*

*ΔlonΔmgrB* and *E. coli ΔmgrB,* respectively. Each point represents fold change value for a single gene. Mean of the scatter is shown as a black line, and its value is provided along with standard deviation above the plot. (**C, D**) Correlation between expression levels of genes in LTMPR1 (**C**) and WTMPR4 (**D**) with *E. coli ΔlonΔmgrB* and *E. coli ΔmgrB,* respectively. Gene expression levels were estimated using RNA-sequencing and fold changes were calculated using wild-type *E. coli* as reference for WTMPR4 and *E. coli ΔmgrB,* and *E. coli Δlon* as reference for LTMPR1 and *E. coli ΔlonΔmgrB*. Each point represents log$_2$(fold change) value for a single gene. The resulting scatter was fit to a simple linear regression and the obtained R$^2$ values are provided. (**E, F**) Relative fitness (w) of indicated *E. coli* strains calculated using competition experiments with *E. coli ΔlacZ* as reference in the absence (**E**) or presence (**F**) of 10 mM MgSO$_4$. Presence of high concentration of Mg$^{2+}$ alleviates the fitness cost of *mgrB* mutations by inhibiting the PhoQP system (*Vinchhi et al., 2023*). For strains derived from *E. coli Δlon*, *E. coli ΔlonΔlacZ* was used as reference. Neutrality of *ΔlacZ* was established by competition with the unmarked ancestral strains, i.e., wild-type (WT) and *E. coli Δlon*. No change in relative fitness compared to the reference strain (w=1) is indicated by a dotted line. Mean ± SD from three independent measurements are plotted. Statistical significance was tested using an unpaired t-test and a p-value of <0.5 was considered statistically significantly different (*). A p-value≥0.5 was considered not significantly different (ns).

The online version of this article includes the following source data and figure supplement(s) for figure 2:

**Source data 1.** Annotated image files for western blots in *Figure 2A*.

**Source data 2.** Raw image files for western blots in *Figure 2A*.

**Figure supplement 1.** Gene expression changes associated with *folA*-encompassing duplication.

fixation. Further, the region of the genome amplified in GDA-2 was similar to what we had observed when *E. coli Δlon* had evolved in drug-free media (*Figure 1—figure supplement 3*, *Figure 3—figure supplement 1*), indicating that GDA-2 was likely a fitness enhancing mutation unrelated to trimethoprim adaptation.

To further investigate the emergence of GDA-2 during evolution of LTMPR1 in drug-free media, we sequenced six randomly picked colonies from each of the lineages at the final time point (*Figure 3F*). All the sequenced isolates had a single copy of *folA* and a majority of isolates showed the presence of GDA-2 (*Figure 3F*, *Supplementary file 2*). Interestingly, some of the isolates that did not harbour GDA-2 had evolved mutations in *rpoS* (*Figure 3F*, *Supplementary file 2*). Loss-of-function mutations in RpoS are fixed at late time points during adaptation to trimethoprim and enhance the fitness of *E. coli* without conferring resistance (*Patel and Matange, 2021*; *Vinchhi et al., 2023*). We also sequenced a few randomly picked isolates from WTMPR4 and WTMPR5 evolved lines (two from each of the three replicate lineages in each case) to check for the presence of GDA-2 and *rpoS* mutations. As expected, clones derived from WTMPR4 had reverted to a single copy of *folA*, while WTMPR5 retained its ancestrally inherited *folA* mutation. None of them had evolved the GDA-2 mutation, while a few of the isolates from WTMPR4 and WTMPR5 lines had acquired *rpoS* mutations (*Figure 3G and H*).

Consistent with earlier studies, and the role of mutations in *rpoS* in general fitness enhancement in *E. coli*, we found that an LTMPR1-derived isolate with mutant *rpoS* recovered fitness in drug-free media (*Figure 4A*). Interestingly, an isolate with the GDA-2 mutation had also recovered fitness (*Figure 4A*). Since GDA-2 occurred only in a *lon*-deficient background, and GDA-2 and *rpoS* mutations were mutually exclusive, we wondered whether the fitness enhancing effects of GDA-2 were also mediated by reduced RpoS activity. Lon is known to reduce the levels of RpoS during the exponential growth phase of *E. coli* (*Ranquet and Gottesman, 2007*). Therefore, we reasoned that at least some targets of RpoS would be overexpressed in a *lon*-knockout. To test this prediction, we analysed the transcriptome of *E. coli Δlon* and specifically looked at the expression levels of 168 known targets of RpoS (*Supplementary file 3*). Of these, 41 genes were overexpressed in the *lon*-knockout by at least 1.5-fold over wild-type, including genes induced in the stationary phase and by various stresses such as *poxB*, *aidB,* and *cbpA* (*Figure 4B and C*). Levels of *rpoS* mRNA were however unaffected in this strain. Next, we compared the expression levels of these 41 RpoS targets in LTMPR1 and *E. coli ΔlonΔmgrB* with *E. coli Δlon* as the reference and found that most had similar expression levels in these three strains (*Figure 4C*). Indeed, a few of these genes showed higher expression levels in LTMPR1 and *E. coli ΔlonΔmgrB*, which was not unexpected since loss of *mgrB* is known to stabilise RpoS (*Xu et al., 2019*). We checked their expression in isolates derived from LTMPR1 which harboured either GDA-2 or *rpoS* mutations. In both isolates RpoS target genes were downregulated, indicating a resetting of RpoS activity (*Figure 4C*). Further, in the isolate harbouring GDA-2, *rpoS* mRNA levels were themselves lowered to 0.49× of *E. coli Δlon* (*Supplementary file 3*). Finally, deletion of *rpoS* from *E. coli Δlon* enhanced the fitness of the strain (*Figure 4D*). Thus, evolution of LTMPR1 in antibiotic-free

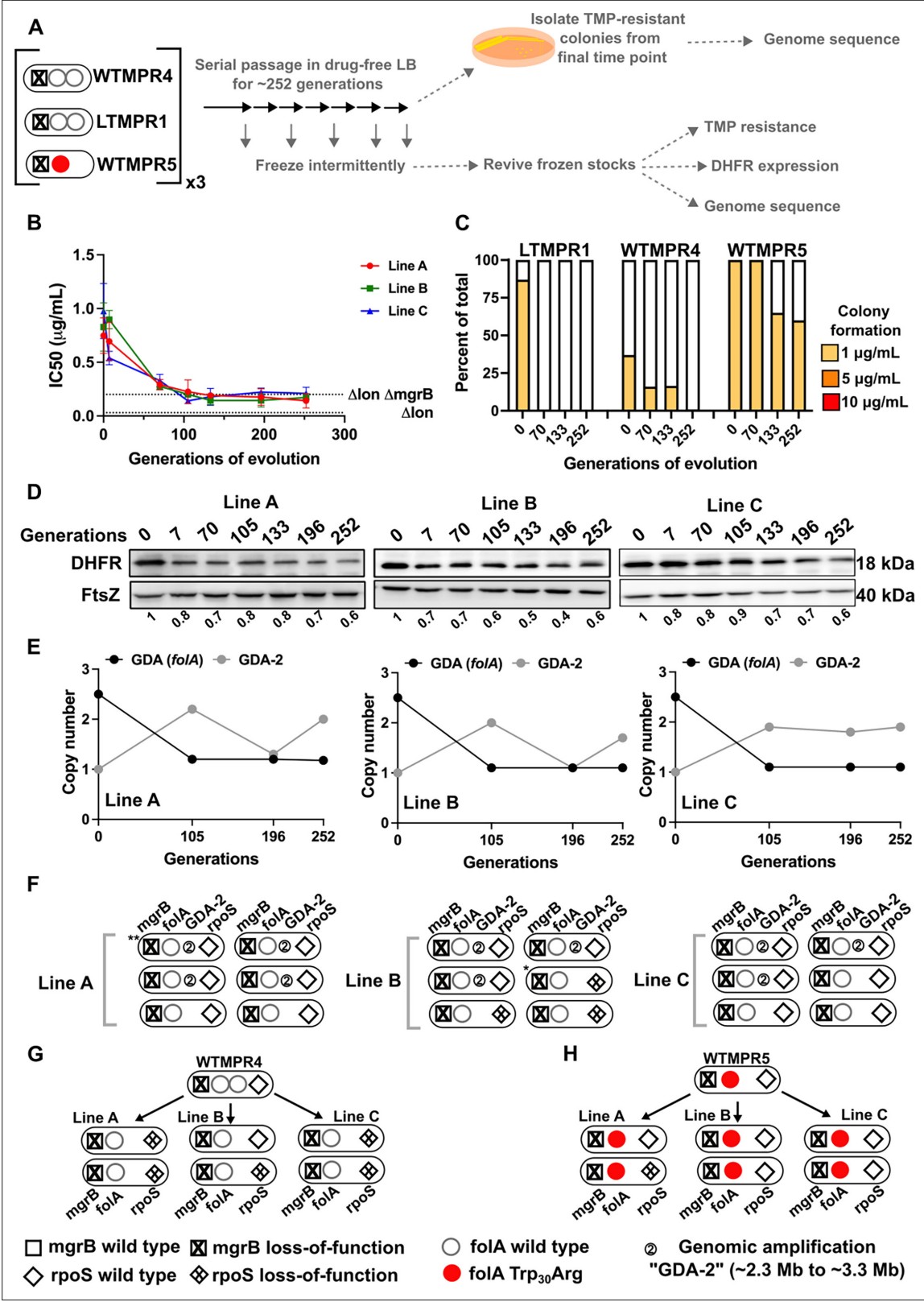

**Figure 3.** Evolutionary fate of *folA* duplication in the absence of drug pressure. (**A**) Schematic of the experimental pipeline used to investigate the impact of evolution in drug-free media on resistance level, *folA* copy number, and expression level of different trimethoprim-resistant *E. coli* populations. (**B**) Trimethoprim inhibitory concentration 50 (IC50) values of three evolving lineages derived from LTMPR1 (**A, B, C**) over 252 generations of evolution in antibiotic-free medium. Mean ± SD from three measurements are plotted at each time point. IC50 values of *E. coli* Δ*lon* and Δ*lon*Δ*mgrB*

*Figure 3 continued on next page*

*Figure 3 continued*

are shown as dotted lines for reference. (**C**) Colony formation of LTMPR1, WTMPR4, and WTMPR5 evolved in antibiotic-free medium at different generations. Fraction of the population capable of forming colonies at 1, 5, and 10 µg/mL trimethoprim was calculated across three replicate lines. Mean value from the three lines is plotted. The results of similar experiments performed on ancestors (0 generations) are also provided for reference. (**D**) Dihydrofolate reductase (DHFR) expression during evolution in the absence of trimethoprim in LTMPR1 lineages **A**, **B**, and **C**, measured by immunoblotting using anti-DHFR polyclonal antibody. FtsZ was used as a loading control. Quantitation was performed by calculating band intensities using image analysis. DHFR expression at each time point was normalised to the ancestor (i.e. 0 generations, set to 1). Mean of three independent measurements is shown below each lane. (**E**) Copy number of the ancestral gene duplication and amplification (GDA) encompassing *folA* (GDA(*folA*)) and GDA-2 at different time points of evolution in the absence of trimethoprim are plotted. For GDA(*folA*), copy number was determined by dividing number of reads from an Illumina sequencing experiment corresponding to *folA* by the average number of reads mapping to the rest of the genome. (**F**) Point mutations in *folA*, *rpoS,* and *mgrB* and the 'GDA-2' genomic duplication in six randomly picked colonies from each of the LTMPR1 lines at 252 generations of evolution are shown schematically using the appropriate symbols. Asterisk (*/**) marks the genotypes that were carried forward for further analyses. (**G, H**) Point mutations in *folA*, *mgrB*, and *rpoS* in isolates derived from 252 generations of evolution of WTMPR4 (**G**) and WTMPR5 (**H**) in no antibiotic. From each of the evolving lines three random isolates were picked for genome sequencing. Various point mutants at the three gene loci of interest are represented by appropriate symbols as shown in the legend.

The online version of this article includes the following source data and figure supplement(s) for figure 3:

**Source data 1.** Annotated image file for western blots in *Figure 3D*.

**Source data 2.** Raw image file for western blots in *Figure 3D*.

**Figure supplement 1.** Coverage depth plots for population sequencing at 105, 196, and 252 generations of the three lineages of LTMPR1 (**A**, **B**, **C**) evolving in trimethoprim-free media.

media was characterised by rapid reversal of GDA-mediated resistance and acquisition of mutations that enhanced fitness by reducing the levels of RpoS-transcribed genes (*Figure 4E*).

## Evolution at constant trimethoprim pressure

In the presence of trimethoprim (300 ng/mL), antibiotic resistance level and DHFR expression remained high over the course of evolution of LTMPR1 (*Figure 5A, B, C, and D*). WTMPR4 and WTMPR5 too remained resistant, and in the case of WTMPR4 an increased colony forming efficiency was observed at a late time point (*Figure 5C*). These data suggested that under drug pressure high *folA* copy number was likely to be maintained. Indeed, population sequencing of LTMPR1 lineages showed that the ancestral GDA mutation was maintained until 196 generations of evolution in all three replicates (*Figure 5E* and *Figure 5—figure supplement 1*). By 252 generations, however, a majority of bacteria in Lines A and C had reverted to a single *folA* copy (*Figure 5E*, *Figure 5—figure supplement 1*) and had acquired a point mutation either in the *folA* gene (Line C) or its promoter (Line A) (*Figure 5E*, *Supplementary file 4*). Line B, on the other hand, retained high *folA* copy number, but with a reduction in the length of the genomic duplication (*Figure 5D*, *Figure 5—figure supplement 1*). Line B had also acquired a mutation in the *folA* promoter, however, its frequency in the population was low (*Figure 5D*, *Supplementary file 4*).

In agreement with population sequencing, none of the resistant isolates derived from LTMPR1 Line A at the final time point showed the ancestral GDA. Instead, they all harboured a $C_{-35}T$ mutation in the *folA* promoter that is known to upregulate DHFR expression (*Vinchhi et al., 2023*; *Figure 5F*, *Supplementary file 4*). Among Line B isolates, many had retained the ancestral duplication, without acquiring any additional mutations in *folA*. One of the isolates from this Line that had reverted to a single *folA* copy had gained the $A_{-64}T$ *folA* promoter mutation (*Figure 5F*, *Supplementary file 4*). In Line C too we found that isolates that had lost the ancestral GDA had evolved a point mutation in the *folA* promoter ($+TG_{-30}$) or a Trp$_{30}$Arg mutation in DHFR (*Figure 5F*, *Supplementary file 4*). We found similar results in sequenced isolates from WTMPR4 lineages, i.e., loss of GDA and gain of a point mutation either in *folA* or its promoter (*Figure 5G*). WTMPR5-derived isolates did not accumulate additional mutations in *folA* but did acquire mutations in *rpoS* (*Figure 5H*). Interestingly, two isolates from LTMPR1 Lines B and C had evolved new GDA mutations encompassing *folA*. In one of the Line B isolates (Isolate 2), the ancestral GDA was replaced by 5× amplification of a shorter genomic region (~12 kb) encompassing *folA* (*Figure 5—figure supplement 2*). Both ends of this newly amplified region were flanked by IS1 elements (*Figure 5—figure supplement 2*). On the other hand, one of the Line C isolates (Isolate 5) had an expanded duplication beyond what was originally found in LTMPR1 (*Figure 5—figure supplement 2*). Also, both copies of *folA* in this isolate coded for

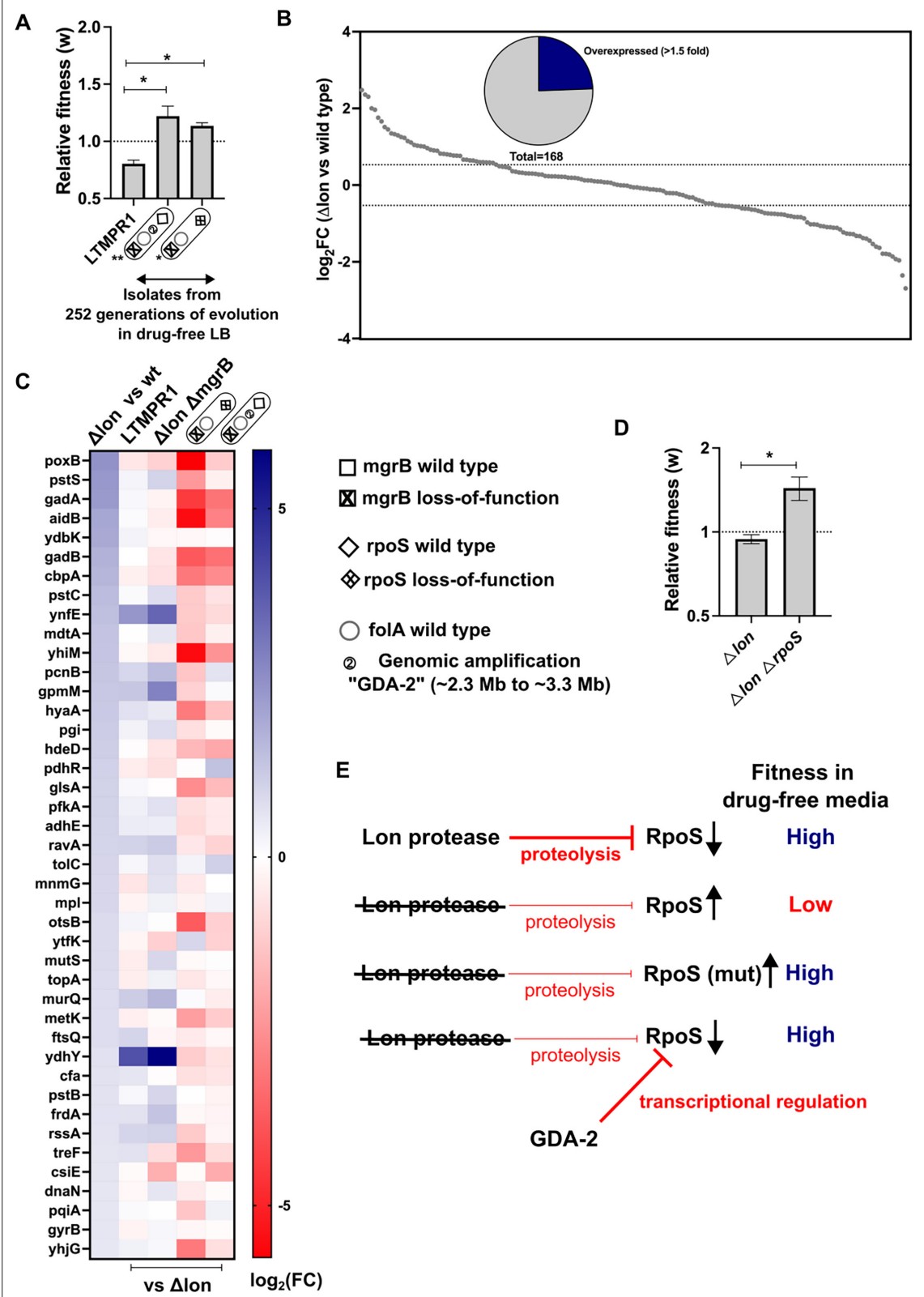

**Figure 4.** Lower RpoS activity leads to fitness enhancement of LTMPR1 in drug-free media. (**A**) Relative fitness (w) of isolates with the indicated genotypes derived from LTMPR1 evolution in antibiotic-free media (see also **Figure 3**) calculated using a competitive growth assay using *E. coli* *ΔlonΔlacZ* as the reference strain. Mean ± SD from three independent measurements are plotted. No change in relative fitness compared to the reference (w=1) is shown as a dotted line for reference. Statistical significance was tested using an unpaired t-test. A p-value of <0.05 was considered

*Figure 4 continued on next page*

*Figure 4 continued*

as a statistically significant difference (\*). (**B**) Expression level of 168 known targets of RpoS in *E. coli Δlon* compared to wild-type expressed as log$_2$(fold change) values determined by RNA-sequencing. Dotted lines at 1.5-fold higher and lower than wild-type represent cutoffs used to identify overexpressed and underexpressed genes. The pie chart shows the fraction of RpoS targets that were overexpressed in *E. coli Δlon* by at least 1.5-fold. (**C**) Expression levels of 41 deregulated RpoS targets in *E. coli Δlon* compared to wild-type are shown as a heat map in the first vertical. The expression levels of these genes in LTMPR1, *E. coli ΔlonΔmgrB*, or LTMPR1-derived isolates compared to *E. coli Δlon* are provided in the subsequent verticals. (**D**) Relative fitness of *E. coli ΔlonΔrpoS* in antibiotic-free media calculated using a competitive growth assay using *E. coli ΔlonΔlacZ* as the reference strain. Neutrality of the *ΔlacZ* genetic marker was verified by competition between *E. coli Δlon* and *E. coli ΔlonΔlacZ*. Mean ± SD from three independent measurements are plotted. No change in relative fitness compared to the reference (w=1) is shown as a dotted line for reference. Statistical significance was tested using an unpaired t-test. A p-value of <0.5 was considered as a statistically significant difference (\*). (**E**) Model for the effects of Lon deficiency on RpoS levels and bacterial fitness. The roles of mutations within RpoS or the gene duplication and amplification (GDA)-2 mutation in compensating for the defects of RpoS overproduction are shown.

DHFR-Trp$_{30}$Arg (**Figure 5E**, **Figure 5—figure supplement 2**), suggesting that the original GDA was lost and an extended GDA mutation was gained after fixation of the mutant *folA* allele.

Based on these results we concluded that even under drug pressure, GDA mutations were reversed in most bacteria. Notably, reversion to a single *folA* copy was slower than in drug-free media as it first required the fixation of point mutations at the *folA* locus. However, in a smaller subpopulation, more than one *folA* gene copy was maintained. This maintenance was dynamic and likely involved repeated loss and gain of GDA mutations.

To further investigate the replacement of GDA by point mutants, we checked to see if point mutants could invade a population harbouring GDA mutations under drug pressure. We allowed LTMPR1-derived isolates with point mutations in *folA* (marked with *ΔlacZ::Cat*) to compete with a GDA-harbouring isolate from the same time point in evolution and asked how rapidly the former could invade a population of the latter. We performed these competitions with a contemporary GDA-harbouring isolate rather than the ancestor to nullify the effects of other media adaptations such as *rpoS* mutations that may have been acquired over evolution. Since for this assay we used point mutant strains genetically marked with *ΔlacZ::Cat* we verified the neutrality of the *ΔlacZ::Cat* marker by checking if the marked isolate could invade its unmarked parent. In the presence of trimethoprim (300 ng/mL), point mutants were enriched over the GDA mutant in coculture by about 100-fold (**Figure 6A**) recreating the results of our evolution experiment. A possible explanation for this finding may be higher resistance level conferred by *folA* point mutations than GDA. Indeed, some isolates with *folA* point mutations had higher IC50 values compared to a contemporary GDA-harbouring isolate (**Figure 6B**). However, regardless of IC50 values, all isolates also had similar doubling times in the presence of 300 ng/mL trimethoprim (**Figure 6B**). Thus, though there were no large observable differences in growth rate between isolates, point mutants could indeed invade and outcompete GDA mutations, reflecting either subtle differences in fitness between them or their relative stabilities in culture.

## Maintenance of high *folA* gene copy number under drug pressure is driven by unstable mutants of DHFR

Though sustained drug pressure did extend the lifetime of *folA* gene duplication, acquisition of trimethoprim-resistant point mutations facilitated reversion to a single gene copy. We therefore asked whether *folA* duplication could be maintained under conditions of increasing antibiotic pressure, i.e., trimethoprim concentrations starting at 1 μg/mL and ending at 10 μg/mL over 252 generations of evolution (**Figure 7A**). Under these conditions, trimethoprim IC50 values increased sharply over the course of evolution eventually reaching ~40- to ~160-fold higher than ancestral LTMPR1 (**Figure 7B**). Colony formation at high trimethoprim concentrations also increased at later generations of evolution (**Figure 7C**). Similarly evolved WTMPR4 and WTMPR5 also showed enhanced colony formation at high trimethoprim concentrations (**Figure 7C**).

Curiously, we observed that the three replicate lineages of LTMPR1 showed different trends in DHFR expression. Line A maintained a high level of DHFR protein for the duration of the experiment. On the other hand, Lines B and C showed high DHFR expression until 196 generations, after which it sharply declined (**Figure 7D**). Genome sequencing revealed that all three lines convergently evolved the Trp$_{30}$Arg mutation in DHFR early during evolution and had fixed this mutation by 196 generations.

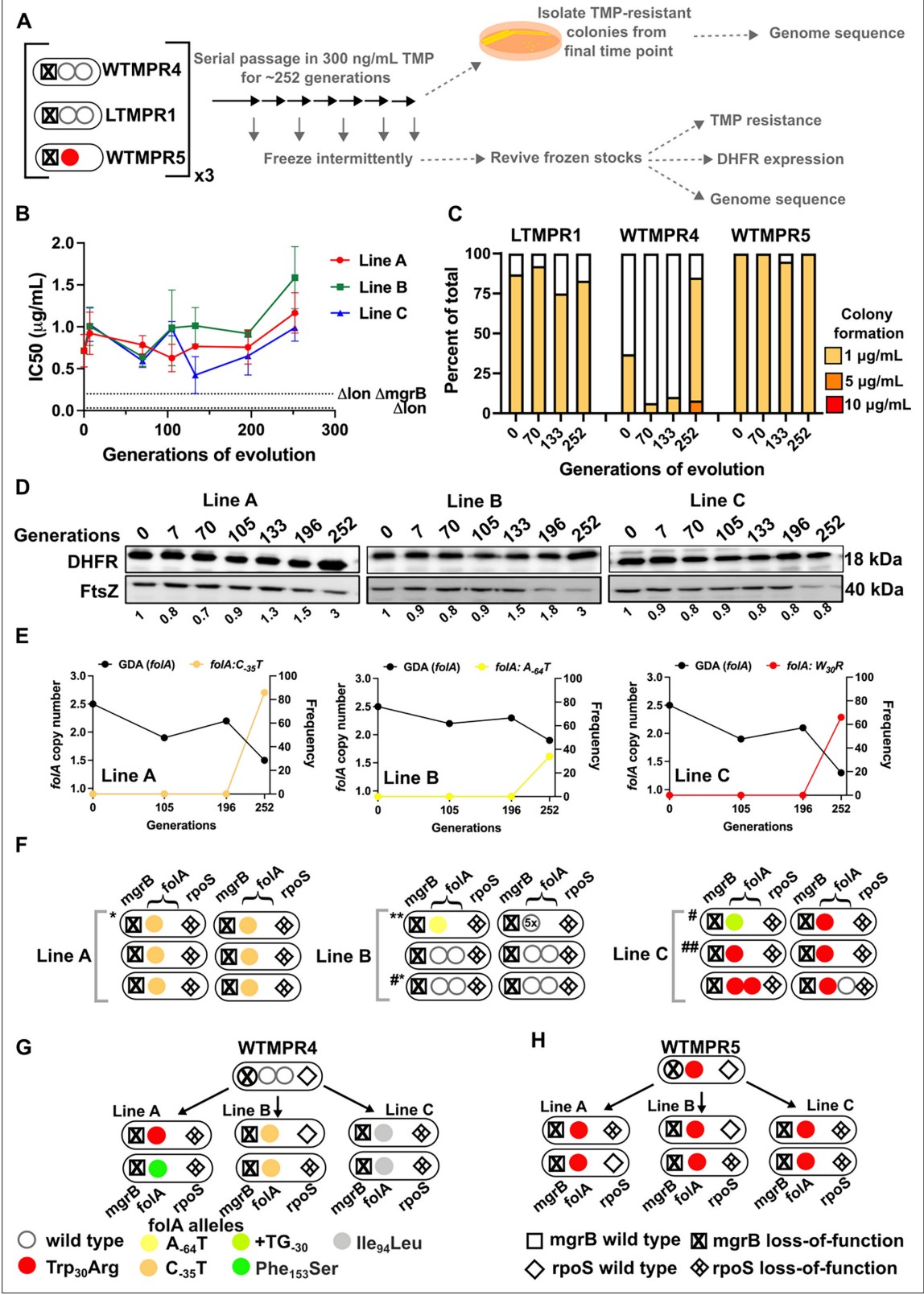

**Figure 5.** Evolutionary fate of *folA*-encompassing gene duplication and amplification (GDA) in LTMPR1 at sustained trimethoprim pressure. (**A**) Schematic of the experimental pipeline used to investigate the impact of evolution in constant drug pressure on resistance level, *folA* copy number, point mutations in *folA,* and expression level of different trimethoprim-resistant *E. coli* populations. (**B**) Trimethoprim inhibitory concentration 50 (IC50) values of three evolving lineages starting from LTMPR1 (A, B, C) over 252 generations of evolution at 300 ng/mL trimethoprim. Mean ± SD from three

*Figure 5 continued on next page*

*Figure 5 continued*

measurements are plotted at each time point. IC50 values of *E. coli* Δ*lon* and Δ*lon*Δ*mgrB* are shown as dotted lines for reference. (**C**) Colony formation of LTMPR1, WTMPR4, and WTMPR5 evolved in 300 ng/mL trimethoprim at indicated time points. Fraction of the population capable of forming colonies at 1, 5, and 10 μg/mL trimethoprim was calculated across three replicate lines. Mean value from the three lines is plotted. The results of similar experiments performed on ancestors (0 generations) are also provided for reference. (**D**) Dihydrofolate reductase (DHFR) expression in LTMPR1 lineages (**A**, **B**, and **C**) evolved in 300 ng/mL trimethoprim, measured by immunoblotting using anti-DHFR polyclonal antibody. FtsZ was used as a loading control. Quantitation was performed by calculating band intensities using image analysis. DHFR expression at each time point was normalised to the ancestor (i.e. 0 generations, set to 1). Mean of three independent measurements is shown below each lane. (**E**) Copy number of *folA* (GDA(*folA*)) at different time points of evolution of LTMPR1 in trimethoprim is plotted on the left y-axis. Copy number was determined by dividing number of reads from an Illumina sequencing experiment corresponding to *folA* by the average number of reads mapping to the rest of the genome. Frequency of various *folA* alleles in the evolving populations at each of the time points is plotted on the right y-axis. (**F**) Point mutations in *folA*, *rpoS,* and *mgrB* in six randomly picked trimethoprim-resistant colonies from each of the LTMPR1 lines at 252 generations of evolution are shown diagrammatically. Genotypes that were carried forward for further analysis are marked * or #. (**G, H**) Point mutations in *folA*, *mgrB*, and *rpoS* in isolates derived from 252 generations of evolution of WTMPR4 (**G**) and WTMPR5 (**H**) in 300 ng/mL of trimethoprim. From each of the evolving lineages two random isolates were picked for genome sequencing. Various point mutants at the three gene loci are represented by appropriate symbols as shown in the legend.

The online version of this article includes the following source data and figure supplement(s) for figure 5:

**Source data 1.** Annotated image file for western blots in *Figure 5D*.

**Source data 2.** Raw image file for western blots in *Figure 5D*.

**Figure supplement 1.** Coverage depth plots for population sequencing at 105, 196, and 252 generations of the three lineages of LTMPR1 (**A, B, C**) evolving in trimethoprim-supplemented (300 ng/mL) media.

**Figure supplement 2.** Coverage plots for Line C Isolate 5 and Line B Isolate 2.

Until this time point, all three lines retained *folA* duplication (*Figure 7E*, *Figure 7—figure supplement 1*). Subsequently, Line A evolved an additional $C_{-35}T$ *folA* promoter mutation and had partially lost the ancestral GDA (*Figure 7E*, *Figure 7—figure supplement 1*, *Supplementary file 5*). In Lines B and C, a second missense mutation in DHFR, i.e., $Tyr_{151}Asp$, appeared between 196 and 252 generations (*Figure 7D*, *Supplementary file 5*). Line B also showed an $Ile_{94}Leu$ mutation in DHFR at low frequency at generation 252 (*Figure 7D*, *Supplementary file 5*). Unexpectedly, despite acquiring additional mutations and lower DHFR expression level, LTMPR1 Lines B and C retained two copies of *folA* in contrast to Line A (*Figure 7E*, *Figure 7—figure supplement 1*, *Supplementary file 5*).

Individual resistant clones from each of the lineages provided greater insight into the genetic trajectories of evolution. In LTMPR1 Line A, two of the six sequenced isolates had lost gene duplication and acquired the $Trp_{30}Arg$ coding and $C_{-35}T$ promoter mutations in *folA* (*Figure 7F*, *Supplementary file 5*). Interestingly, the other four sequenced isolates from Line A retained the ancestral GDA and harboured the same combination of *folA* point mutations, but only in one of the two gene copies (*Figure 7F*, *Supplementary file 5*). Ten of the twelve isolates from LTMPR1 Lines B and C, on the other hand, retained *folA* duplication. In all 10 isolates, both copies of *folA* had acquired $Trp_{30}Arg$ and $Tyr_{151}Asp$ mutations (*Figure 7F*, *Supplementary file 5*). Unlike LTMPR1, all sequenced isolates of WTMPR4 at 252 generations of evolution had lost their GDA mutation and gained two *folA* mutations (*Figure 7G*). WTMPR5 too had gained an additional *folA* mutation (*Figure 7H*). Notably, none of the isolates from WTMPR4 and WTMPR5 lineages had the $Trp_{30}Arg$ and $Tyr_{151}Asp$ combination of mutations that occurred repeatedly in LTMPR1 lineages.

We next sought to understand why some LTMPR1 bacteria had retained high *folA* copy number despite acquiring multiple point mutations in the drug target. To address this question, we first asked what the advantage of *folA* duplication was in the two different mutational trajectories. Isolates from Line A had similar IC50 values for trimethoprim regardless of whether they had one or two copies of *folA* (*Figure 8A*). On the contrary, isolates from Line B that harboured only one copy of mutant *folA* (coding for DHFR-$Trp_{30}Arg$-$Tyr_{151}Asp$) displayed ~3.5-fold lower IC50 than those that harboured two copies (*Figure 8A*). Immunoblotting showed that isolates from Line A had similar expression level of DHFR (*Figure 8B*). However, DHFR protein level was significantly compromised by the $Trp_{30}Arg$-$Tyr_{151}Asp$ mutation combination and double the number of *folA* gene copies restored this defect (*Figure 8B*). Thus, higher gene dosage of DHFR was dispensable once a promoter upregulatory mutation was fixed. On the other hand, bacteria with alleles of DHFR harbouring multiple missense mutations continued to benefit from overexpression due to *folA* duplication.

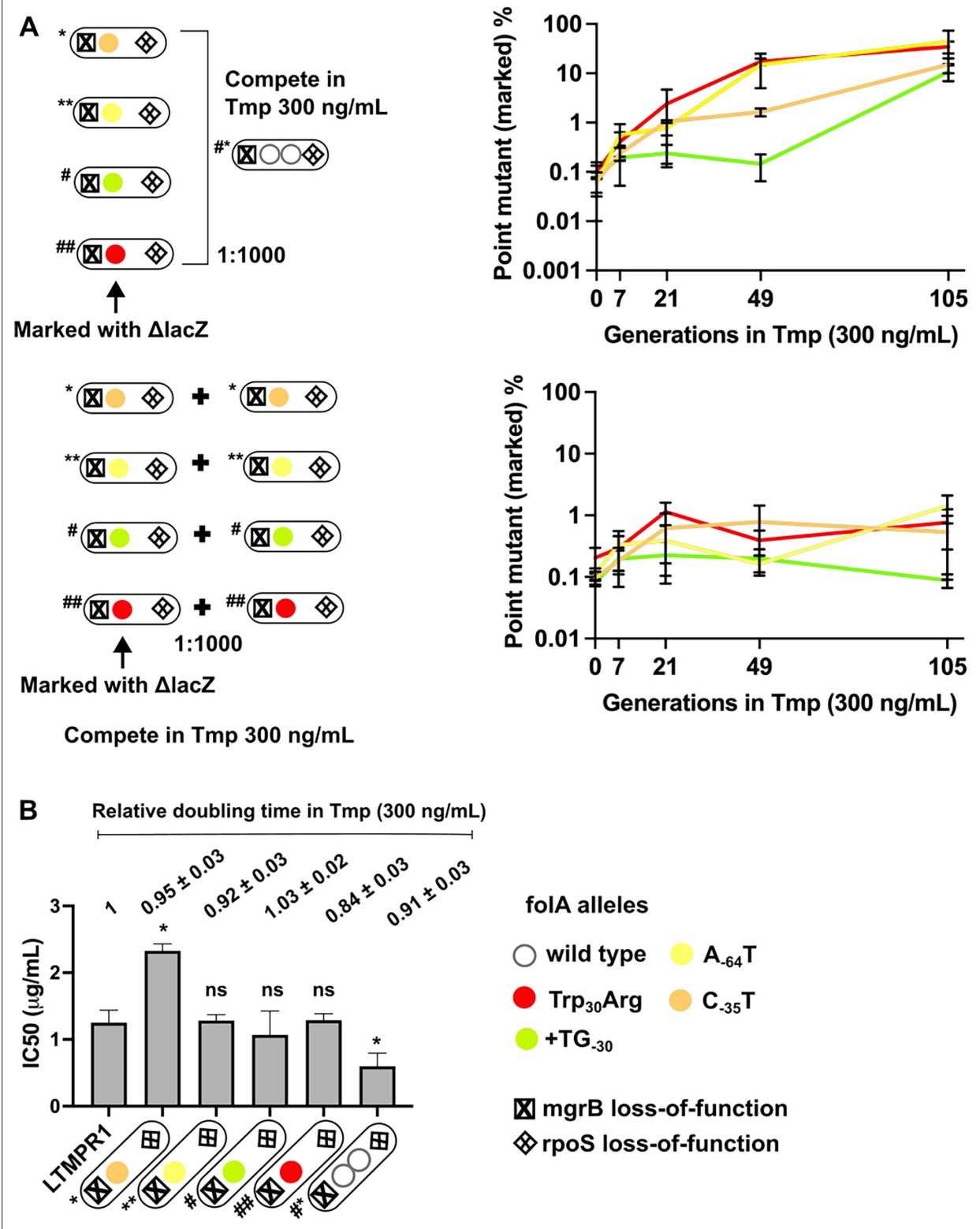

**Figure 6.** Invasion of gene duplication and amplification (GDA)-dominant populations by point mutations under trimethoprim pressure. (**A**) Competition between isolates with point mutations in *folA* and an isolate that harboured a *folA* duplication derived from the same time point of LTMPR1 evolution (upper panel). The genotypes of the isolates (i.e. alleles at *mgrB*, *folA,* and *rpoS* loci) used are shown diagrammatically (see also *Figure 5*, *Supplementary file 4*). The initial mixing ratio was 1000:1 in favour of the GDA mutant. Point mutants were marked genetically with *ΔlacZ::Cat.* Neutrality of the *ΔlacZ::Cat* marker was verified by competing marked and unmarked point mutants (lower panel). Percentage of the population

*Figure 6 continued on next page*

*Figure 6 continued*

constituted by the marked point mutants over ~105 generations of serial transfer in media supplemented with trimethoprim (300 ng/mL) is plotted. Mean ± SD from three replicates are plotted and traces for individual mutants are appropriately coloured. (**B**) Trimethoprim inhibitory concentration 50 (IC50) values of the LTMPR1 ancestor or evolved isolates from 252 generations in trimethoprim with indicated genotypes (see also *Figure 5*, *Supplementary file 4*) are plotted as bars. Mean ± SD from three independent measurements are plotted. Statistical significance was tested using an unpaired t-test and LTMPR1 was used as the reference. A p-value<0.05 was considered significantly different (*) while ≥0.5 was considered not statistically significant (ns). Doubling times relative to LTMPR1 (set to 1) in the presence of 300 ng/mL of trimethoprim are provided above the graph as mean ± SD from three independent measurements.

Mutations at position $Trp_{30}$ in DHFR render the protein resistant to inhibition by trimethoprim but are accompanied by lower in vivo proteolytic stability due to perturbation of the hydrophobic core of the protein (*Matange et al., 2018*). This affects the expression level of DHFR, resulting in lower cellular abundance that limits the resistance level conferred by mutations at $Trp_{30}$ (*Matange, 2020*; *Matange et al., 2018*). Interestingly, residue $Tyr_{151}$ lies in the same hydrophobic pocket as $Trp_{30}$ (*Figure 8C*). To test the effect of combination of mutations at these sites on trimethoprim resistance and DHFR expression, we first heterologously expressed wild-type DHFR, $Trp_{30}Arg$, or $Tyr_{151}Asp$ single mutants and the $Trp_{30}Arg$-$Tyr_{151}Asp$ double mutant in wild-type *E. coli* (*Figure 8D*). As expected, DHFR $Trp_{30}Arg$ expression increased the IC50 of trimethoprim. Surprisingly, neither DHFR $Tyr_{151}Asp$ nor the $Trp_{30}Arg$-$Tyr_{151}Asp$ double mutant conferred resistance to the antibiotic (*Figure 8E*). Indeed, both mutants conferred lower IC50 values than wild-type DHFR. This effect was due to loss of hydrophobicity at the $Tyr_{151}$ position, as DHFR-$Tyr_{151}Leu$ and DHFR-$Tyr_{151}Phe$ conferred similar IC50 values as wild-type DHFR (*Figure 8E*). Since the $Trp_{30}Arg$-$Tyr_{151}Asp$ mutation combination had evolved in a *lon*-deficient background, we next expressed these mutants in *E. coli Δlon* (*Figure 8D*). Here too, DHFR $Trp_{30}Arg$ conferred trimethoprim resistance, while the $Tyr_{151}Asp$ mutant did not (*Figure 8E*). Strikingly, the combination of $Trp_{30}Arg$ and $Tyr_{151}Asp$ conferred very high-level trimethoprim resistance (*Figure 8E*), demonstrating synergistic epistasis that was contingent on the absence of Lon protease. The levels of expression of DHFR mutants in wild-type and *lon*-deficient bacteria mirrored trimethoprim resistance levels indicating that clearance by Lon altered their genotype-phenotype relationship (*Figure 8F*). Based on these data, we concluded that the action of proteostatic machinery generated additional demand for expression by clearing mutant-DHFR proteins. This additional demand drove the maintenance of *folA* duplications in bacteria evolving at very high drug pressures.

## Discussion

Ohno's hypothesis, proposed in 1970, provided a conceptual framework to think about the evolution and eventual fate of duplicated genes (*Ohno, 1970*). Several theoretical and experimental studies have built upon the original idea to delineate the evolutionary outcomes of gene duplication (*Innan and Kondrashov, 2010*; *Bergthorsson et al., 2007*; *Long et al., 2013*). More recently, experimental evolution of microbes has been used to test some of the suggestions made by Ohno and others. This approach has the inherent advantage of greater control over the selection pressure applied and availability of genetic tools to interrogate molecular mechanisms. In most studies, an artificial system was constructed and then evolved under defined selection pressures (*Tomanek et al., 2020*; *Holloway et al., 2007*; *Kugelberg et al., 2006*; *Naseeb et al., 2017*; *Tomanek and Guet, 2022*). Our study began with serendipitous isolation of spontaneous trimethoprim-resistant mutants of *E. coli* that overproduced DHFR because of a *folA* gene duplication. We exploited this system to understand the relationship between selection pressure and gene duplication in the context of drug resistance evolution. The selection pressure for higher gene expression, i.e., 'expression demand', can be understood as the amount of a gene product that is required to maintain fitness in a particular environment. Antibiotics directly increase expression demand by inhibiting the activity of their targets. Our study shows that increasing gene dosage is a rapid, but transient evolutionary response to ensure that expression demand is met. On the other hand, point mutations are relatively stable routes to mitigating expression demand. Over short timescales, changes in gene dosage can counter the effects of antibiotic. However, their inherent instability results in almost inevitable replacement with phenotypically equivalent point mutations. Thus, continued antibiotic exposure delayed but could not prevent reversion to low gene copy number in most of our experiments. Further, we discover a novel association between

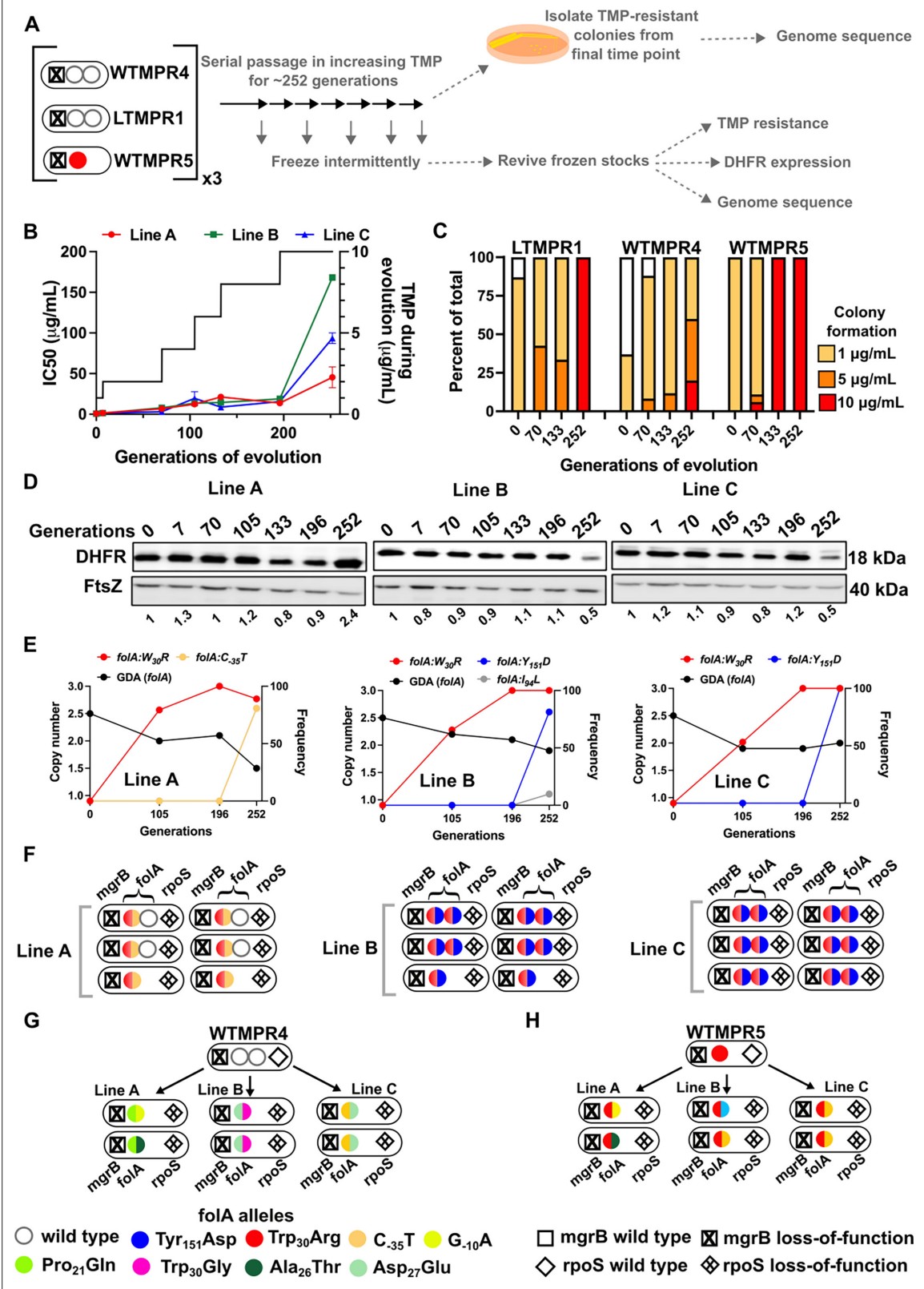

**Figure 7.** Evolutionary fate of *folA*-encompassing gene duplication and amplification (GDA) in LTMPR1 at increasing trimethoprim pressure. (**A**) Schematic of the experimental pipeline used to investigate the impact of evolution in increasing drug pressure on resistance level, *folA* copy number, point mutations in *folA*, and expression level of different trimethoprim-resistant *E. coli* populations. (**B**) Trimethoprim inhibitory concentration 50 (IC50) values of three evolving lineages starting from LTMPR1 (**A, B, C**) over 252 generations of evolution are plotted on the left y-axis. Mean ± SD from three

*Figure 7 continued on next page*

*Figure 7 continued*

measurements are plotted at each time point. Trimethoprim concentrations used during evolution are plotted in the right y-axis. (**C**) Colony formation of LTMPR1, WTMPR4, and WTMPR5 evolved in increasing trimethoprim at indicated time points. Fraction of the population capable of forming colonies at 1, 5, and 10 μg/mL trimethoprim was calculated across three replicate lines. Mean value from the three lines is plotted. The results of similar experiments performed on ancestors (0 generations) are also provided for reference. (**D**) Dihydrofolate reductase (DHFR) expression in LTMPR1 lineages **A**, **B**, and **C** evolved in increasing trimethoprim, measured by immunoblotting using anti-DHFR polyclonal antibody. FtsZ was used as a loading control. Quantitation was performed by calculating band intensities using image analysis. DHFR expression at each time point was normalised to the ancestor (i.e. 0 generations, set to 1). Mean of three independent measurements is shown below each lane. (**E**) Copy number of *folA* (GDA(*folA*)) at different time points of LTMPR1 evolution in increasing trimethoprim is plotted on the left y-axis. Copy number was determined by dividing number of reads from an Illumina sequencing experiment corresponding to *folA* by the average number of reads mapping to the rest of the genome. Frequency of various *folA* alleles in the evolving LTMPR1 populations at each of the time points is shown on the right y-axis. (**F**) Point mutations in *folA*, *rpoS*, and *mgrB* in six randomly picked trimethoprim-resistant colonies from each of the LTMPR1 lines at 252 generations of evolution are shown schematically. (**G, H**) Point mutations in *folA*, *mgrB*, and *rpoS* in isolates derived from 252 generations of evolution of WTMPR4 (**G**) and WTMPR5 (**H**) in increasing trimethoprim. From each of the evolving lineages two random isolates were picked for genome sequencing. Various point mutants at the three gene loci are represented by appropriate symbols as shown in the legend.

The online version of this article includes the following source data and figure supplement(s) for figure 7:

**Source data 1.** Annotated image file for western blots in *Figure 7D*.

**Source data 2.** Raw image file for western blots in *Figure 7D*.

**Figure supplement 1.** Coverage depth plots for population sequencing at 105, 196, and 252 generations of the three lineages of LTMPR1 (**A, B, C**) evolving in increasing trimethoprim concentrations.

---

proteostasis and gene dosage evolution. In *E. coli* evolving at high trimethoprim pressure, bacterial protein quality control exacerbated expression demand by clearing mutant DHFR-s. As a result, drug-resistant mutants of DHFR with low proteolytic stability drove the maintenance of *folA* gene duplication (*Figure 9*). Based on these findings we propose that a combination of factors, which include environmental and physiological, alter the relationship between gene copy number and fitness, and drive gene dosage evolution.

A key finding from our study was the twofold influence that proteostatic machinery had on gene copy number. Firstly, Lon protease deficiency promoted GDA mutations, possibly by allowing accumulation of IS-tranposase proteins in *E. coli*. This observation resonates with earlier work by Nicoloff and coworkers who found that large IS-element flanked genomic duplications in tetracycline-resistant *E. coli* were commonly preceded by spontaneous mutations that inactivated *lon* (*Nicoloff et al., 2007*; *Nicoloff et al., 2006*). Further, the role of Lon protease in regulating IS-element mobility by a similar mechanism has been previously demonstrated for IS1 and IS903 (*Derbyshire et al., 1990*; *Rouquette et al., 2004*). We note however that other mechanisms by which Lon may modulate the frequency of IS-mediated GDAs in *E. coli* cannot be ruled out. For instance, Lon protease homologs from several organisms directly bind DNA (*Hua et al., 2020*; *Karlowicz et al., 2017*; *Liu et al., 2004*; *Lu et al., 2007*; *Sonezaki et al., 1995*), though the role of this binding in DNA transactions is unclear. Similarly, Lon is a regulator of the recombination protein, RecA, by its action on LexA (*Breidenstein et al., 2012*) and in *Salmonella* RecA stimulated the rate of gene duplications mediated by IS3 (*Reams et al., 2012*). The second impact of protein quality control on GDA was the action of proteases on trimethoprim-resistant mutants of DHFR, which promoted gene duplication by generating additional demand for expression (*Figure 9*). Importantly, which of these two effects dominated was determined by a combination of environment, i.e., drug concentrations, and other mutations acquired at the *folA* locus during evolution. In particular, mutations in the *folA* promoter could alleviate expression demand and hence reduce selection for high gene copy number.

The interplay between GDA and point mutations during evolution that we report here agrees in many respects with a recent study by *Tomanek and Guet, 2022*. They use an elegantly designed artificial system (*Tomanek et al., 2020*; *Tomanek and Guet, 2022*) in which expression of the *galK* gene in *E. coli* was driven by a very low activity promoter. This strain was then allowed to evolve in different concentrations of galactose. Like the present study, they found that the activity of IS-elements biased evolutionary trajectories in favour of duplications. They also showed that co-occurrence of duplications and point mutations was selected only at high expression demand, in agreement with our findings. Finally, they argued that gene duplication events may constrain evolution by point mutations. Our study corroborates these results at the short term in that the *lon*-knockout which showed a higher

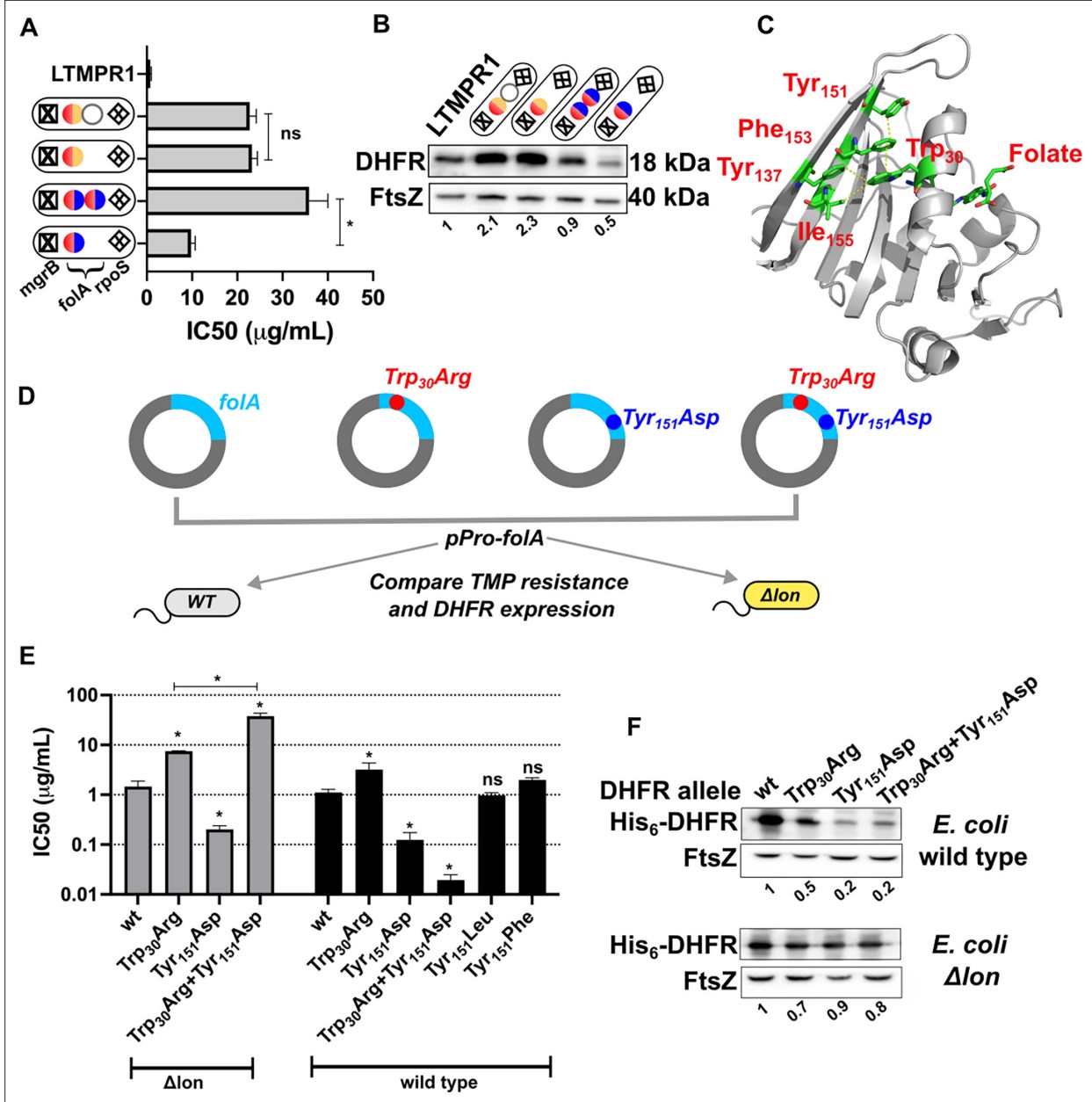

**Figure 8.** Proteostatic pressure facilitates maintenance of *folA* duplication. (**A**) Trimethoprim inhibitory concentration 50 (IC50) values of resistant isolates with the indicated genotypes (see also *Figure 7*) derived from LTMPR1 evolution in increasing antibiotic pressure. Mean ± SD from three independent measurements are plotted. Statistical significance was tested using an unpaired t-test. A p-value of <0.05 was considered statistically significant (*). A p-value>0.05 was considered not statistically significant (ns). (**B**) Dihydrofolate reductase (DHFR) expression level in LTMPR1-derived trimethoprim-resistant isolates with indicated genotypes assessed by immunoblotting using anti-DHFR polyclonal antibody. FtsZ was used as a loading control. Quantitation was performed by calculating band intensities using image analysis. DHFR expression was normalised to the ancestor (LTMPR1). Mean of three independent measurements is provided. (**C**) Cartoon representation of the structure of *E. coli* DHFR (PDB: 7DFR) bound to folate. Residues Trp$_{30}$, Tyr$_{151}$, Phe$_{153}$, and Ile$_{155}$ which form hydrophobic interactions and are required for proteolytic stability are shown as sticks and coloured by element (C: green, O: red, N: blue). Distances of less than or equal to 4 Å, which indicates possible interactions, are shown as dotted yellow traces. Folate bound in the active site of DHFR is shown as sticks. (**D**) Schematic of the experiment used to test trimethoprim resistance and expression level of various DHFR mutants in *E. coli* wild-type (WT) or *Δlon*. (**E**) Trimethoprim IC50 values of *E. coli* wild-type or *E. coli Δlon* heterologously expressing DHFR (wt) or its mutants. Mean ± SD from three independent measurements are plotted. Statistical significance was tested using an unpaired t-test. Comparisons were between mutant and wild-type DHFR, unless otherwise indicated. A p-value of <0.05 was considered statistically significant (*). A p-value>0.05 was considered not statistically significant (ns). (**F**) Expression level of plasmid-borne DHFR (wt) or its mutants in wild-type or *Δlon E. coli* assessed by immunoblotting using anti-DHFR polyclonal antibody. FtsZ was used as a loading control. Quantitation was performed by calculating band intensities using image analysis. Expression of mutants was normalised to wild-type DHFR (set to 1). Mean of three independent measurements is shown.

*Figure 8 continued on next page*

*Figure 8 continued*

The online version of this article includes the following source data for figure 8:

**Source data 1.** Annotated image file for western blots in *Figure 8B and F*.

**Source data 2.** Raw image file for western blots in *Figure 8B and F*.

propensity for GDA mutations did not evolve point mutations in *folA* at early time points of evolution. However, given the relative low stability of large GDAs, these were inevitably lost over longer duration of evolution and replaced with point mutations. Thus, it is likely that the different mutational profiles of GDA-biased populations may be relevant only at short durations of evolution. It is important to note that though we have focused on the effects of mutations at the *folA* locus, which are directly under trimethoprim pressure, many of the isolates from our evolution experiments also harboured additional mutations at other loci. For instance, mutations in *rpoS* were seen throughout our evolution lines and at least in drug-free media were significant contributors to bacterial fitness. It is not yet clear what the

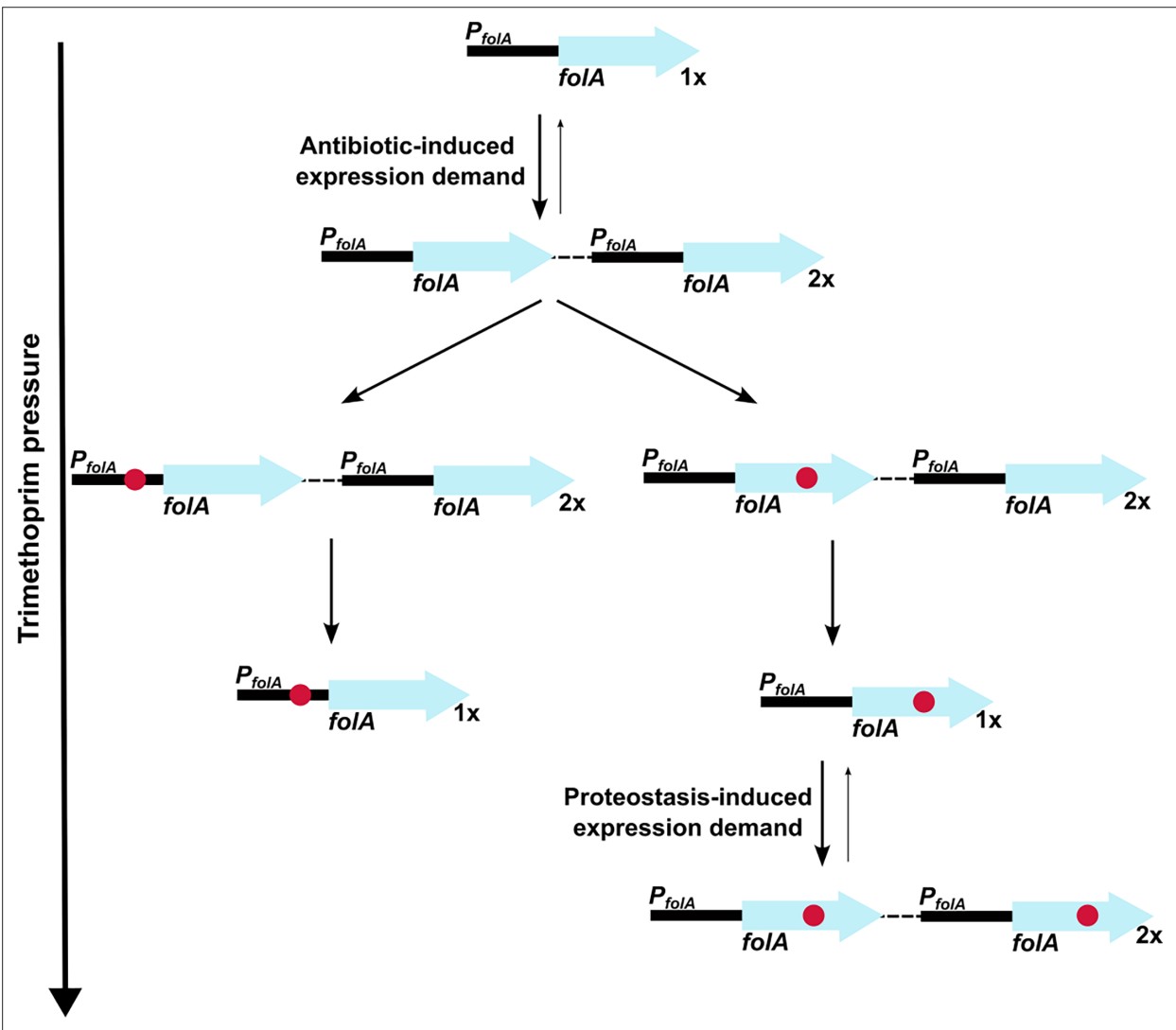

**Figure 9.** Model of expression demand induced selection of *folA* gene duplication, followed by replacement by phenotypically equivalent point mutants. Expression demand for dihydrofolate reductase (DHFR) is generated by trimethoprim pressure, which results in the selection of *folA* gene duplications. Gene duplication is inherently unstable and reverses to a single copy of wild-type *folA* when drug pressure is withdrawn. Under drug pressure, *folA* gene duplication is maintained until a point mutation that confers resistance arises in one of the copies of *folA* (promoter or coding region). Upon acquisition of a point mutation, reversal of gene duplication ensues unless additional expression demand is generated by the action of proteostatic machinery on unstable drug-resistant DHFR mutants.

role of other mutations are likely to be under antibiotic selection, though it may be envisaged that these additional mutations play the role of media adaptations. We did not take their effects into this study account since we expected them to be minor contributors to fitness, especially during evolution under trimethoprim selection. However, their role in modulating the stability and selection of GDA mutations in drug-free/low drug environments may be more significant and remains to be investigated in the future.

In summary, the present study uncovers the complex relationship between gene copy number, point mutations, and environment, and identifies proteostasis as a regulator of gene dosage evolution in bacteria. These findings open up the possibility of alternative explanations for how gene duplications may be maintained over long periods of evolution and provides insight into how gene expression demand is mitigated by multiple mechanisms in bacteria. Moreover, our findings relating proteostasis in general and Lon in particular, to adaptive mutations add to a growing body of literature that define a causal link between protein quality control and mutational paths of evolution. For DHFR itself, decreasing the stringency of protein quality control changes the fitness landscape of single mutations and their combinations, altering the mutational repertoires accessible during adaptive evolution (*Matange, 2020*; *Rodrigues et al., 2016*; *Guerrero et al., 2019*; *Thompson et al., 2020*). Similar effects of protein homeostatic machinery on mutant fitness are reported from a diverse range of proteins that influence phenomena like viral evolution (*Yoon et al., 2023*) and the fate of horizontally acquired genes (*Bershtein et al., 2015*; *Brown et al., 2010*; *Standley et al., 2022*). Indeed, a central role for protein quality control machinery in molecular and organismal evolution is reflected in conserved signatures of the proteostatic network across the different kingdoms of life (*Koutsandreas et al., 2024*). These links may be important to consider for a new wave of potential antibacterials that target proteostasis (*Khodaparast et al., 2023*; *Lupoli et al., 2018*; *Won et al., 2024*). It appears critical to ensure that the impact of changes in evolvability are considered when designing such strategies for therapeutic intervention.

# Materials and methods

**Key resources table**

| Reagent type (species) or resource | Designation | Source or reference | Identifiers | Additional information |
|---|---|---|---|---|
| Strain, strain background (*Escherichia coli*) | *E. coli* K-12 MG1655 | Gift from Prof. Manjula Reddy, CCMB, India | NCBI:txid511145 | |
| Antibody | Anti-DHFR (Rabbit polyclonal) | *Matange et al., 2018* | | (100 ng/mL) |
| Antibody | Anti-FtsZ (Rabbit polyclonal) | Gift from Prof. Manjula Reddy, CCMB, India | | (1:50,000) |
| Recombinant DNA reagent | pPRO-*folA* | *Matange et al., 2018* | Plasmid for expression of DHFR | |
| Chemical compound, drug | Trimethoprim | Sigma-Aldrich | Cat. No. T7883 | |

## Strains, plasmids, and culture conditions

*E. coli* K-12 MG1655 was used as wild-type for all experiments. *E. coli* DH10B was used for plasmid construction and manipulation. *E. coli* strains were cultured in Luria-Bertani (LB) broth with shaking at 180–200 rpm or on LB agar plates at 37°C. Ampicillin was used at a concentration of 100 µg/mL for selection of plasmid-transformed strains. Trimethoprim was added to growth media at required concentrations as needed. The strains and plasmids used in this study are listed in *Table 2*.

## Laboratory evolution of antibiotic resistance

### Short-term evolution of antibiotic resistance

For short-term laboratory evolution of antibiotic resistance, wild-type and *E. coli* Δ*lon* populations were cultured in 96-well plates in LB medium incorporating subinhibitory concentrations of different antibiotics. Following concentrations of antibiotics were used: trimethoprim (300 ng/mL; MIC/3); spectinomycin (5 µg/mL; MIC/10); nalidixic acid (4.25 µg/mL; MIC/3); erythromycin (10 µg/mL; MIC/10); rifampicin (4.25 µg/mL; MIC/3). Control populations were also set up in antibiotic-free LB. Five replicates were included for each strain in each antibiotic. Total volume of culture in each well was 150 µL.

**Table 2.** List of strains and plasmids used in the study.

| Sr. no. | Strain | Description | Source |
|---|---|---|---|
| 1. | *Escherichia coli* K-12 MG1655 | Wild-type *E. coli*; used as ancestor for evolution experiments and as reference strains for phenotypic and genotypic comparisons | Laboratory stocks |
| *Gene knockouts generated in E. coli K-12 MG1655* | | | |
| 1. | *E. coli ΔmgrB* | Isogenic knockout of *mgrB* gene (*ΔmgrB::Kan*) | *Patel and Matange, 2021* |
| 2. | *E. coli ΔlacZ* | Isogenic knockout of *lacZ* gene (*ΔlacZ::Cat*) | *Patel and Matange, 2021* |
| 3. | *E. coli Δlon* | Isogenic knockout of *lon* gene (*Δlon::Kan*) | *Matange, 2020* |
| 4. | *E. coli ΔlonΔlacZ* | Double knockout of *lon* (*Δlon::Kan*) and *lacZ* (*ΔlacZ::Cat*) genes | *Patel and Matange, 2021* |
| 5. | *E. coli ΔmgrBΔlon* | Double knockout of *lon* (*Δlon::Cat*) and *mgrB* (*ΔmgrB::Kan*) genes | *Patel and Matange, 2021* |
| 6. | *E. coli ΔlonΔrpoS* | Double knockout of *lon* (*Δlon::Cat*) and *rpoS* (*ΔrpoS::Kan*) genes | *Patel and Matange, 2021* |
| 7. | *E. coli ΔpitA* | Isogenic knockout of *pitA* gene (*ΔpitA::Kan*) | This study |
| *Strains/isolates derived from short-term adaptive laboratory evolution* | | | |
| 1. | WTMPR1, WTMPR2, WTMPR3, WTMPR4, and WTMPR5 | Trimethoprim-resistant derivatives of *E. coli* wild-type obtained by laboratory evolution at 300 ng/mL of trimethoprim for ~25 generation, followed by selection on agar plates supplemented with 1 µg/mL of trimethoprim | *Patel and Matange, 2021*. Note: These strains are labelled as TMPR1–5 in the original publication. They have been renamed WTMPR1-5 to indicate that they were derived after evolution of wild-type. |
| 2. | LTMPR1, LTMPR2, LTMPR3, LTMPR4, and LTMPR5 | Trimethoprim-resistant derivatives of *E. coli Δlon* obtained by laboratory evolution at 300 ng/mL of trimethoprim for ~25 generation, followed by selection on agar plates supplemented with 1 µg/mL of trimethoprim | This study |
| *Plasmids* | | | |
| 1. | pPRO-*insB* | *InsB* gene cloned into the pPROEx-HtC expression plasmid in frame with an N-terminal hexa-histidine tag | This study |
| 2. | pPRO-*insL* | *InsL* gene cloned into the pPROEx-HtA expression plasmid in frame with an N-terminal hexa-histidine tag | This study |
| 3. | pPRO-*folA* | *FolA* gene cloned into the pPROEx-HtB expression plasmid in frame with an N-terminal hexa-histidine tag | *Matange et al., 2018* |
| 4. | pPRO-*folA* Trp$_{30}$Arg | *FolA* gene cloned into the pPROEx-HtB expression plasmid, harbouring the Trp$_{30}$Arg mutation in frame with an N-terminal hexa-histidine tag | *Matange et al., 2018* |
| 5. | pPRO-*folA* Tyr$_{151}$Asp | *FolA* gene cloned into the pPROEx-HtB expression plasmid, harbouring the Tyr$_{151}$Asp mutation in frame with an N-terminal hexa-histidine tag | This study |
| 6. | pPRO-*folA* Trp$_{30}$Arg Tyr$_{151}$Asp | *FolA* gene cloned into the pPROEx-HtB expression plasmid, harbouring the Trp$_{30}$Arg and Tyr$_{151}$Asp mutations in frame with an N-terminal hexa-histidine tag | This study |
| 7. | pPRO-*folA* Tyr$_{151}$Phe | *FolA* gene cloned into the pPROEx-HtB expression plasmid, harbouring the Tyr$_{151}$Phe mutation in frame with an N-terminal hexa-histidine tag | This study |
| 8. | pPRO-*folA* Tyr$_{151}$Leu | *FolA* gene cloned into the pPROEx-HtB expression plasmid, harbouring the Tyr$_{151}$Leu mutation in frame with an N-terminal hexa-histidine tag | This study |

10% of the population was passaged to fresh growth media every 24 hr, for 7 days, resulting in ~25 generations of evolution. At the end of 7 days, the evolved populations were frozen as glycerol stocks.

To check for the evolution of resistance and isolation of resistant clones for further analyses, frozen populations were revived in drug-free LB broth and grown overnight (14–16 hr). Saturated cultures were serially diluted and spotted on LB agar plates supplemented with the appropriate antibiotic at MIC. Colonies were randomly picked, cultured overnight in drug-free LB, and frozen in 50% glycerol at –80°C until further use.

## Long-term evolution at different trimethoprim pressures

For long-term evolution, trimethoprim-resistant isolates LTMPR1, WTMPR4, and WTMPR5 were first revived from frozen stocks in drug-free LB overnight. Triplicate evolving lineages were then established by subculturing 1% of the saturated culture of each ancestor into 2 mL of appropriate medium. For relaxed selection drug-free LB was used. For constant antibiotic pressure, trimethoprim was added at a concentration of 300 ng/mL. For increasing antibiotic pressure, trimethoprim was increased from 1 to 10 µg/mL over course of the experiment as shown in *Figure 7B*. Cultures were grown for 10–14 hr (~7 generations) and then passaged (1%) into fresh media. This was continued for 36 passages, i.e., ~252 generations. Aliquots of cultures were frozen periodically for further analyses.

For isolating individual clones, populations were revived in drug-free LB from frozen stocks at 252 generations of evolution in drug-free LB, serially diluted and then spotted onto LB agar plates supplemented with trimethoprim (1, 5, or 10 µg/mL). Plates were incubated for 18–20 hr at 37°C and resistant colonies were picked randomly for further analyses.

## Measuring trimethoprim resistance

### Trimethoprim IC50

Trimethoprim IC50, i.e., the antibiotic concentration needed for 50% growth inhibition, was calculated using a broth dilution assay. Frozen glycerol stocks of appropriate bacterial strains were revived in 1–3 mL LB for 6–8 hr. In a 96-well plate, 150 µL of LB was added to individual wells. Trimethoprim stocks were added, and the antibiotic was serially diluted. No antibiotic was added to a few wells, which served as controls. Next, 1.5 µL of the appropriate bacterial culture was added to each well. The peripheral wells of the plate were filled with water to prevent dehydration. The 96-well plate was then incubated at 37°C for 15–18 hr. Optical density at 600 nm was measured for each well using a plate reader (Ensight, Perkin Elmer). Obtained data were first normalised to growth in drug-free wells (set to 1) and then fitted to a sigmoidal log(inhibitor)-response curve using GraphPad Prism software. The equation for this model is as follows:

$$Y = \text{Min} + (\text{Max} - \text{Min})/(1 + 10^{(X - \text{Log(IC50)})}).$$

where Min: Minimum value of normalised OD and Max: maximum value of normalised OD.

Values of IC50 were estimated from the above curve fit and averaged across three independent measurements for every strain.

### Colony forming efficiency

To calculate the colony forming efficiency of evolving populations at different trimethoprim concentrations, frozen glycerol stocks were first revived in 1 mL LB for 14–15 hr. Saturated cultures were serially diluted (10-fold). Diluted cultures were then spotted (10 mL) on LB agar plates containing 0, 1, 5, and 10 µg/mL of trimethoprim. Spots were allowed to dry completely and the plates were incubated at 37°C for ~18 hr. Colonies were counted from the least dense spot and multiplied by the appropriate dilution factor to estimate colony forming units per mL (CFUs/mL) of culture.

### Growth curves

Appropriate frozen stocks were revived overnight in drug-free LB (1–3 mL) and 1 µL of the saturated culture was added to 200 µL of media supplemented with trimethoprim (300 ng/mL) in the wells of a sterile uncoated 96-well plate. Optical density at 600 nm was measured during growth at 37°C at 15 min intervals over a 24 hr period on a Perkin-Elmer Ensight Multimode Plate Reader. Doubling time (DT) (in min) was estimated from log phase, using the following formula:

$$DT = \ln(2) / \left( \frac{\ln\left(\frac{OD2}{OD1}\right)}{T2 - T1} \right)$$

where OD1 and OD2 represent optical densities at the onset and end of log phase, respectively, and T1 and T2 represent time elapsed (in min), corresponding to OD1 and OD2.

## Genome sequencing, identification of mutations, and detection of GDA mutations

Genomic DNA was extracted from late log phase cultures of appropriate strain/resistant isolates/evolving populations using phenol:chloroform:isoamyl alcohol (25:24:1) extraction as previously described (*Matange et al., 2019*). Extracted genomic DNA was then cleaned up using HiGenoMB DNA isolation and purification kit (HiMedia, India) as per the manufacturer's instructions and quantitated spectrophotometrically. The integrity was checked by running purified DNA on an agarose gel before further analyses.

For whole-genome resequencing, paired-end whole-genome next-generation sequencing was performed on a MiSeq System (Illumina, USA) with read lengths of 150–250 bp. Library preparation and sequencing services were provided by Eurofins (Bangalore, India) and Genotypic (Bangalore, India). Processed reads were aligned to the reference genome *E. coli* K-12 MG1655 (U00096) using bowtie2. All sequenced genomes had an average coverage depth of at least 100×. Variant calling and prediction of new junctions in the genome to identify large structural mutations was done using Breseq (*Deatherage and Barrick, 2014*) using default settings in population or clonal modes as needed. All variants and new junctions that were already present in the ancestor were excluded. Variants unique to the test sample were classified as 'mutations'.

For detection of GDA mutations, 'coverage' vs 'coordinate in reference genome' plot generated by the bam.cov command in Breseq was used (*Deatherage and Barrick, 2014*). Contiguous stretches greater than 10 kb with coverage depth of 1.5× or more, compared to the rest of the genome, were classified as GDA mutations. The presence of IS-elements or repeat sequences was manually checked using the coordinates of the GDA.

## Calculating relative fitness of *E. coli* strains

Relative fitness of gene knockouts or evolved clones was estimated using competition assays. Saturated cultures of test strains and reference strains (*E. coli ΔlacZ* or *E. coli ΔlacZ Δlon*) were mixed 1:1 to a total of 1% in 3 mL LB. Strains were allowed to compete for ~7 generations (24 hr incubation). Mixed cultures were diluted and plated on LB agar supplemented with IPTG (1 mM) and X-gal (50 μg/mL) at the start and end of the competition experiment. Plates were incubated for 18–24 hr, and blue and white colonies were counted to determine the CFUs/mL at initial and final time points.

Relative fitness (w) was calculated using the following formula:

$$w = \frac{\ln\left(\frac{Tf}{Ti}\right)}{\ln\left(\frac{Rf}{Ri}\right)}$$

where $T_f$ and $T_i$ are the final and initial CFUs/mL of the test strain and $R_f$ and $R_i$ are the final and initial CFUs/mL of the reference strain, respectively.

## Estimation of DHFR expression by immunoblotting

Immunoblotting was used to determine the expression level of DHFR in various strains/mutants of *E. coli* as described in *Patel and Matange, 2021*. Briefly, equal lysate protein (5 μg) was subjected to SDS-PAGE on a 15% gel and electroblotted onto PVDF membranes. The membrane was blocked with 5% BSA solution and cut into two halves at the 25 kDa molecular weight band. The lower part of the membrane was probed with anti-DHFR polyclonal IgG (100 ng/mL) and the upper part was probed with anti-FtsZ serum (1:50,000). HRP-linked anti-rabbit IgG (1:10,000) was used as the secondary antibody. Hybridised antibodies were detected using chemiluminescence. Band intensities were quantitated in ImageJ and normalised to appropriate control (set to 1). Fold change in DHFR was calculated as follows:

$$Fold\ change_{DHFR} = \frac{Normalised\ intensity_{DHFR}}{Normalised\ intensity_{FtsZ}}$$

## Gene expression analysis using RNA-sequencing

Transcriptome analyses were performed using RNA-sequencing. For RNA-sequencing, saturated overnight culture of appropriate strains/isolates was inoculated (1%) into 3 mL LB in triplicate and incubated at 37°C for 3 hr with shaking at 180 rpm. The cultures were centrifuged and bacterial pellets were resuspended in 1 mL of RNAlater (Invitrogen) for storage. RNA extraction, sequencing, and preliminary data analyses were performed by Eurofins (Bangalore, India). Total RNA was extracted using RNAeasy spin columns and quantitated using Qubit and Bioanalyzer. Bacterial rRNA was depleted using Ribozero kit. Bacterial mRNA was then reverse-transcribed, library prepared, and quality control performed using TapeStation platform. Paired-end sequencing was performed on Illumina platform with read lengths of 150 bp. Processed reads were aligned to the reference genome (U00096.3) using HISTAT2 (version 2.1.0). Abundance estimation was done using featureCounts (version 1.34.0). Differential gene expression (DGE) analysis was done by comparing the expression of individual genes in test and reference samples. The output of the DGE analyses were $\log_2$(fold change) values for each gene and p-values for statistical significance.

## Cloning and site-directed mutagenesis

### Cloning of InsB and InsL transposases

*InsB* was amplified from *E. coli* K-12 MG1655 genomic DNA by PCR using primers insB1_fwd_*Nco*I (5'-GGCGCCATGGATATGCCGGGCAACAGCCCG-3') insB1_rev_*Eco*RI (5'- CCTTTGAATTCTTATTGATAGTGTTTTATG-3'). For amplification of *insL*, primers insL_*Eco*RI_fwd (5'- ATGCCGAATTCATGAATTACTCTCACGATAAC-3') and insL_*Hin*dIII_rev (5'- ATGCCAAGCTTTTAGTTCTTCTTTTCGGATCC-3') were used. Amplified *insB* gene was digested with *Nco*I and *Eco*RI restriction endonuclease and ligated with similarly digested pPROEx-HtC plasmid. Amplified *insL* was digested with *Eco*RI and *Hin*dIII restriction endonucleases and ligated with similarly digested pPROEX-HTa plasmid. Positive clones were screened using restriction digestion. Clones were transformed into *E. coli* wild-type or Δ*lon* for further analyses.

### Site-directed mutagenesis for DHFR

Site-directed mutagenesis was performed to introduce mutations into the coding sequence of *folA* in pPRO-folA as described in *Matange et al., 2018*. Briefly, PCR was performed using pPRO-folA or pPRO-folA *Trp$_{30}$Arg* as template and the appropriate forward and reverse mutagenic primers given below:

> folA_*Tyr$_{151}$Asp*_fwd_*Bgl*II:
> 5'-CTCTCACAGCGATTGCTTTGAGATCTTGGAGCGGCGGGGCC-3' folA_*Tyr$_{151}$Asp*_rev_*Bgl*II:
> 5'- CGCC GCTC CAAG ATCT CAAA GCAA TCGC TGTG AGAG TTCT GCG-3'
> folA_*Tyr$_{151}$Leu*_fwd_*Bgl*II:
> 5'-TCTCACAGCTTATGCTTTGAGATCTTGGAGCGGCGGGGCC-3' folA_*Tyr$_{151}$Leu*_rev_*Bgl*II:
> 5'-                    CGCCGCTCCAAGATCTCAAAGCATAAGCTGTGAGAGTTCTGCG-3'
> folA_*Tyr$_{151}$Phe*_fwd_*Bgl*II:
> 5'- CTCACAGCTTTTGCTTTGAGATCTTGGAGCGGCGGGGCC-3' folA_*Tyr$_{151}$Phe*_rev_*Bgl*II:
> 5'- GCCGCTCCAAGATCTCAAAGCAAAAGCTGTGAGAGTTCTGCG-3'

PCR products were digested with *Dpn*I overnight, transformed into chemically competent *E. coli* DH10B and selected on LB agar plates containing ampicillin. Plasmids habouring the desired mutations were screened by restriction digestion and confirmed by sequencing. Sequenced clones were transformed into *E. coli* wild-type or Δ*lon* for further analyses.

## In-cell stability of InsB and InsL

A chloramphenicol chase assay was used to assess the in-cell stability of heterologously expressed InsB and InsL transposases as described in *Matange, 2020*, with a few modifications. Overnight cultures of *E. coli* wild-type or Δ*lon* harbouring plasmids pPRO-insB or pPRO-insL were diluted 1% in 5 mL LB supplemented with ampicillin and IPTG (0.1 mM). After 4 hr of growth at 37°C, 1 mL of culture was pelleted. The cell pellet was lysed in 100 μL of 1× Laemmli sample loading buffer by heating at 95°C for 10–20 min. Chloramphenicol was added to the remaining culture at a final concentration of 25 μg/mL to stop protein synthesis and the culture was placed back in the incubator. After 15, 45, and

90 min, 1 mL aliquots of the culture were pelleted and lysed. Lysates were subjected to SDS-PAGE followed by immunoblotting to detect the levels of InsB and InsL. Anti-Hexahis monoclonal antibody (Invitrogen) was used as the primary antibody for InsL. Anti-Hexahis polyclonal antibody (Invitrogen) was used as the primary antibody for InsB. Band intensities were calculated in ImageJ and normalised to control (set to 1) to calculate fraction of protein remaining after chloramphenicol treatment.

## Material availability statement

All strains, mutants, and plasmids generated will be shared upon request.

## Acknowledgements

We acknowledge Dr. Manjula Reddy for providing us with the anti-FtsZ polyclonal antibody. We acknowledge Bruhad Dave for his valuable inputs during discussions at early stages of this project and Prof. Satyajit Rath, Prof. Sandhya Visweswariah, and Dr. Girish Deshpande for their comments on the manuscript. Funding for this work was provided by DBT/Wellcome Trust India Alliance. NM is a recipient of the India Alliance Intermediate Fellowship. CJ is a recipient of a Senior Research Fellowship from the Council of Scientific and Industrial Research (CSIR), Government of India.

## Additional information

### Funding

| Funder | Grant reference number | Author |
| --- | --- | --- |
| Wellcome Trust/DBT India Alliance | IA/I/20/2/505181 | Nishad Matange |

The funders had no role in study design, data collection and interpretation, or the decision to submit the work for publication. For the purpose of Open Access, the authors have applied a CC BY public copyright license to any Author Accepted Manuscript version arising from this submission.

### Author contributions

Chinmaya Jena, Investigation, Visualization, Methodology, Writing – review and editing; Saillesh Chinnaraj, Investigation, Visualization; Soham Deolankar, Investigation; Nishad Matange, Conceptualization, Resources, Data curation, Formal analysis, Supervision, Funding acquisition, Validation, Investigation, Visualization, Methodology, Writing – original draft, Project administration, Writing – review and editing

### Author ORCIDs

Chinmaya Jena ⓘ https://orcid.org/0000-0002-4401-3570
Nishad Matange ⓘ https://orcid.org/0000-0002-2947-3931

Reviewer #1 (Public review): https://doi.org/10.7554/eLife.99785.3.sa1
Reviewer #3 (Public review): https://doi.org/10.7554/eLife.99785.3.sa2
Author response https://doi.org/10.7554/eLife.99785.3.sa3

## Additional files

### Supplementary files

Supplementary file 1. Gene expression changes in LTMPR1 and WTMPR4 isolates.

Supplementary file 2. Genomic changes associated with evolution of LTMPR1 in drug-free media.

Supplementary file 3. Gene expression changes in isolates derived from LTMPR1 evolution in drug-free media containing mutation in *rpoS* or gene duplication and amplification (GDA)-2.

Supplementary file 4. Genomic changes associated with evolution of LTMPR1 in trimethoprim (300 ng/mL).

Supplementary file 5. Genomic changes associated with evolution of LTMPR1 in increasing trimethoprim pressure.

Supplementary file 6. Genome sequencing of isolates derived from WTMPR4 evolution at different trimethoprim pressures.

Supplementary file 7. Genome sequencing of isolates derived from WTMPR5 evolution at different trimethoprim pressures.

MDAR checklist

## Data availability

Genome and RNA sequencing data have been submitted under accession codes PRJNA1108937, PRJNA741586 and PRJNA996735.

The following datasets were generated:

| Author(s) | Year | Dataset title | Dataset URL | Database and Identifier |
|---|---|---|---|---|
| Matange N | 2024 | Evolution of folA gene duplication under trimethoprim pressure | https://www.ncbi.nlm.nih.gov/bioproject/PRJNA1108937 | NCBI BioProject, PRJNA1108937 |
| Matange N | 2023 | Transcriptome of trimethoprim-resistant bacteria | https://www.ncbi.nlm.nih.gov/bioproject/PRJNA996735 | NCBI BioProject, PRJNA996735 |
| Matange N | 2021 | Trimethoprim resistance in *E. coli* | https://www.ncbi.nlm.nih.gov/bioproject/PRJNA741586 | NCBI BioProject, PRJNA741586 |

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
