## [Editor Report · eLife Assessment]

This **important** study explores the interplay between gene dosage and gene mutations in the evolution of antibiotic resistance. The authors provide **compelling** evidence connecting proteostasis with gene duplication during experimental evolution in a model system. This paper is likely to be of interest to researchers studying antibiotic resistance, proteostasis, and bacterial evolution.

---

## [Referee Report · Reviewer #1 (Public review)]

Summary:

The study by Jena et al. addresses important questions on the fundamental mechanisms of genetic adaptation, specifically, does adaptation proceed via changes of copy number (gene duplication and amplification "GDA") or by point mutation. While this question has been worked on (for example by Tomanek and Guet) the authors add several important aspects relating to resistance against antibiotics and they clarify the ability of Lon protease to reduce duplication formation (previous work was more indirect).

A key finding Jena et al. present is that point mutations after significant competition displace GDA. A second one is that alternative GDA constantly arise and displace each other (see work on GDA-2 in Figure 3). Finally, the authors found epistasis between resistance allele that was contingent on lon. Together this shows an intricate interplay of lon proteolysis for the evolution and maintenance of antibiotic resistance by gene duplication.

Strengths:

The study has several important strengths: (i) the work on GDA stability and competition of GDA with point mutations is a very promising area of research and the authors contribute new aspects to it, (ii) rigorous experimentation, (iii) very clearly written introduction and discussion sections. To me, the best part of the data is that deletion of lon stimulates GDA, which has not been shown with such clarity until now.

Weaknesses:

Previously raised minor weaknesses and technical questions have been adequately resolved in the revised manuscript. As the experiments and their results are described in great detail the interested reader needs stamina. The details will, however, be informative to the specialist.

---

## [Referee Report · Reviewer #3 (Public review)]

Summary:

This is an important paper that investigates the relationship between proteolytic stability of an antibiotic target enzyme and the evolution of antibiotic resistance via increased gene copy number. The target of the antibiotic trimethoprim is dihydrofolate reductase (DHFR). In *Escherichia coli*, DHFR is encoded by folA and the major proteolysis housekeeping protease is Lon (lon). In this manuscript, the authors report the result of the experimental evolution of a lon mutant strain of *E. coli* in response to sub-inhibitory concentrations of the antibiotic trimethoprim then investigate the relationship between proteolytic stability of DHFR mutants and the evolution of folA gene duplication. After 25 generations of serial passaging in a fixed concentration of trimethoprim, the authors found that folA duplication events were more common during evolution of the lon strain, than the wt strain. However, with continued passaging, some folA duplications were replaced by a single copy of folA containing a trimethoprim resistance-conferring point mutation. Interestingly, evolution of the lon strain in the setting of increasing concentrations of trimethoprim resulted in evolved strains with different levels of DHFR expression. In particular, some strains maintained two copies of a mutant folA that encoded an unstable DHFR. In a lon+ background, this mutant folA did not express well and did not confer trimethoprim resistance. However, in the lon- background, it displayed higher expression and conferred high-level trimethoprim resistance. The authors concluded that maintenance of the gene duplication event (and the absence of Lon) compensated for the proteolytic instability of this mutant DHFR. In summary, they provide evidence that the proteolytic stability of an antibiotic target protein is an important determinant of the evolution of target gene copy number in the setting of antibiotic selection.

Strengths:

The major strength of this paper is identifying an example of antibiotic resistance evolution that illustrates the interplay between the proteolytic stability and copy number of an antibiotic target in the setting of antibiotic selection. The results are rigorous and convincingly support the conclusions. This paper will be of interest to any biologist that studies the evolution of resistance mechanisms or gene duplication.

Weaknesses:

The impact of this finding is somewhat limited given that it is a single example that occurred in a lon strain of *E. coli*. Although the specific mechanism is unlikely to occur naturally, this study represents an important and convincing proof of the principle that gene duplication can provide increased expression demand for an unstable resistance determinant in the setting of antibiotic selection.

---

## [Author Response]

The following is the authors’ response to the original reviews.

**Public Reviews:**

**Reviewer #1 (Public review):**
Summary:The study by Jena et al. addresses important questions on the fundamental mechanisms of genetic adaptation, specifically, does adaptation proceed via changes of copy number (gene duplication and amplification "GDA") or by point mutation. While this question has been worked on (for example by Tomanek and Guet) the authors add several important aspects relating to resistance against antibiotics and they clarify the ability of Lon protease to reduce duplication formation (previous work was more indirect).A key finding Jena et al. present is that point mutations after significant competition displace GDA. A second one is that alternative GDA constantly arise and displace each other (see work on GDA-2 in Figure 3). Finally, the authors found epistasis between resistance alleles that was contingent on lon. Together this shows an intricate interplay of lon proteolysis for the evolution and maintenance of antibiotic resistance by gene duplication.Strengths:The study has several important strengths: (i) the work on GDA stability and competition of GDA with point mutations is a very promising area of research and the authors contribute new aspects to it, (ii) rigorous experimentation, (iii) very clearly written introduction and discussion sections. To me, the best part of the data is that deletion of lon stimulates GDA, which has not been shown with such clarity until now.Weaknesses:The minor weaknesses of the manuscript are a lack of clarity in parts of the results section (Point 1) and the methods (Point 2).

We thank the reviewer for their comments and suggestions on our manuscript. We also appreciate the succinct summary of primary findings that the Reviewer has taken cognisance of in their assessment, in particular the association of the Lon protease with the propensity for GDAs as well as its impact on their eventual fate. We have now revised the manuscript for greater clarity as suggested by Reviewer #1.

**Reviewer #2 (Public review):**
Summary:In this strong study, the authors provide robust evidence for the role of proteostasis genes in the evolution of antimicrobial resistance, and moreover, for stabilizing the proteome in light of gene duplication events.Strengths:This strong study offers an important interaction between findings involving GDA, proteostasis, experimental evolution, protein evolution, and antimicrobial resistance. Overall, I found the study to be relatively well-grounded in each of these literatures, with experiments that spoke to potential concerns from each arena. For example, the literature on proteostasis and evolution is a growing one that includes organisms (even micro-organisms) of various sorts. One of my initial concerns involved whether the authors properly tested the mechanistic bases for the rule of Lon in promoting duplication events. The authors assuaged my concern with a set of assays (Figure 8).More broadly, the study does a nice job of demonstrating the agility of molecular evolution, with responsible explanations for the findings: gene duplications are a quick-fix, but can be out-competed relative to their mutational counterparts. Without Lon protease to keep the proteome stable, the cell allows for less stable solutions to the problem of antibiotic resistance.The study does what any bold and ambitious study should: it contains large claims and uses multiple sorts of evidence to test those claims.Weaknesses:While the general argument and conclusion are clear, this paper is written for a bacterial genetics audience that is familiar with the manner of bacterial experimental evolution. From the language to the visuals, the paper is written in a boutique fashion. The figures are even difficult for me - someone very familiar with proteostasis - to understand. I don't know if this is the fault of the authors or the modern culture of publishing (where figures are increasingly packed with information and hard to decipher), but I found the figures hard to follow with the captions. But let me also consider that the problem might be mine, and so I do not want to unfairly criticize the authors.For a generalist journal, more could be done to make this study clear, and in particular, to connect to the greater community of proteostasis researchers. I think this study needs a schematic diagram that outlines exactly what was accomplished here, at the beginning. Diagrams like this are especially important for studies like this one that offer a clear and direct set of findings, but conduct many different sorts of tests to get there. I recommend developing a visual abstract that would orient the readers to the work that has been done.

The reviewer’s comments regarding data presentation are well-taken. Since we already had a diagrammatic model that sums up the chief findings of our study (Figure 9), we have now provided schematics in Figures 1, 3, 5 and 8 to clarify the workflow of smaller sections of the study. We hope that these diagrams provide greater clarity with regards to the experiments we have conducted.

Next, I will make some more specific suggestions. In general, this study is well done and rigorous, but doesn't adequately address a growing literature that examines how proteostasis machinery influences molecular evolution in bacteria.While this paper might properly test the authors' claims about protein quality control and evolution, the paper does not engage a growing literature in this arena and is generally not very strong on the use of evolutionary theory. I recognize that this is not the aim of the paper, however, and I do not question the authors' authority on the topic. My thoughts here are less about the invocation of theory in evolution (which can be verbose and not relevant), and more about engagement with a growing literature in this very area.The authors mention Rodrigues 2016, but there are many other studies that should be engaged when discussing the interaction between protein quality control and evolution.A 2015 study demonstrated how proteostasis machinery can act as a barrier to the usage of novel genes: Bershtein, S., Serohijos, A. W., Bhattacharyya, S., Manhart, M., Choi, J. M., Mu, W., ... & Shakhnovich, E. I. (2015). Protein homeostasis imposes a barrier to functional integration of horizontally transferred genes in bacteria. PLoS genetics, 11(10), e1005612A 2019 study examined how Lon deletion influenced resistance mutations in DHFR specifically: Guerrero RF, Scarpino SV, Rodrigues JV, Hartl DL, Ogbunugafor CB. The proteostasis environment shapes higher-order epistasis operating on antibiotic resistance. Genetics. 2019 Jun 1;212(2):565-75.A 2020 study did something similar: Thompson, Samuel, et al. "Altered expression of a quality control protease in *E. coli* reshapes the in vivo mutational landscape of a model enzyme." Elife 9 (2020): e53476.And there's a new review (preprint) on this very topic that speaks directly to the various ways proteostasis shapes molecular evolution:Arenas, Carolina Diaz, Maristella Alvarez, Robert H. Wilson, Eugene I. Shakhnovich, C. Brandon Ogbunugafor, and C. Brandon Ogbunugafor. "Proteostasis is a master modulator of molecular evolution in bacteria."I am not simply attempting to list studies that should be cited, but rather, this study needs to be better situated in the contemporary discussion on how protein quality control is shaping evolution. This study adds to this list and is a unique and important contribution. However, the findings can be better summarized within the context of the current state of the field. This should be relatively easy to implement.

We thank the reviewer for their encouraging assessment of our manuscript as well as this important critique regarding the context of other published work that relates proteostasis and molecular evolution. Indeed, this was a particularly difficult aspect for us given the different kinds of literature that were needed to make sense of our study. We have now added the references suggested by the reviewer as well as others to the manuscript. We have also added a paragraph in the discussion section (Lines 463-476) that address this aspect and hopefully fill the lacuna that the reviewer points out in this comment.

**Reviewer #3 (Public review):**
Summary:This paper investigates the relationship between the proteolytic stability of an antibiotic target enzyme and the evolution of antibiotic resistance via increased gene copy number. The target of the antibiotic trimethoprim is dihydrofolate reductase (DHFR). In *Escherichia coli*, DHFR is encoded by folA and the major proteolysis housekeeping protease is Lon (lon). In this manuscript, the authors report the results of the experimental evolution of a lon mutant strain of *E. coli* in response to sub-inhibitory concentrations of the antibiotic trimethoprim and then investigate the relationship between proteolytic stability of DHFR mutants and the evolution of folA gene duplication. After 25 generations of serial passaging in a fixed concentration of trimethoprim, the authors found that folA duplication events were more common during the evolution of the lon strain, than the wt strain. However, with continued passaging, some folA duplications were replaced by a single copy of folA containing a trimethoprim resistance-conferring point mutation. Interestingly, the evolution of the lon strain in the setting of increasing concentrations of trimethoprim resulted in evolved strains with different levels of DHFR expression. In particular, some strains maintained two copies of a mutant folA that encoded an unstable DHFR. In a lon+ background, this mutant folA did not express well and did not confer trimethoprim resistance. However, in the lon- background, it displayed higher expression and conferred high-level trimethoprim resistance. The authors concluded that maintenance of the gene duplication event (and the absence of Lon) compensated for the proteolytic instability of this mutant DHFR. In summary, they provide evidence that the proteolytic stability of an antibiotic target protein is an important determinant of the evolution of target gene copy number in the setting of antibiotic selection.Strengths:The major strength of this paper is identifying an example of antibiotic resistance evolution that illustrates the interplay between the proteolytic stability and copy number of an antibiotic target in the setting of antibiotic selection. If the weaknesses are addressed, then this paper will be of interest to microbiologists who study the evolution of antibiotic resistance.Weaknesses:Although the proposed mechanism is highly plausible and consistent with the data presented, the analysis of the experiments supporting the claim is incomplete and requires more rigor and reproducibility. The impact of this finding is somewhat limited given that it is a single example that occurred in a lon strain and compensatory mutations for evolved antibiotic resistance mechanisms are described. In this case, it is not clear that there is a functional difference between the evolution of copy number versus any other mechanism that meets a requirement for increased "expression demand" (e.g. promoter mutations that increase expression and protein stabilizing mutations).

We thank the reviewer for their in-depth assessment of our work and appreciate their concerns regarding reproducibility and rigor in analysis of our data. We have now incorporated this feedback and provided necessary clarifications/corrections in the revised version of our manuscript.

**Recommendations for the authors:**

**Reviewer #1 (Recommendations for the authors):**
Major Points:(1) The authors show that a deletion of lon increases the ability for GDA and they argue that this is adaptive during TMP treatment because it increases the dosage of folA (L. 129). However, the highest frequency of GDA occurred in drug-free conditions (see Figure 1C). This indicates either that GDA is selected in drug-free media and potentially selected against by certain antibiotics. It would help for the authors to discuss this possibility more clearly.

We thank the reviewer for this astute observation. It is indeed striking that the GDA mutation (i.e. the GDA-2 mutation) selected in a lon-deficient background does not come up in presence of antibiotics. To probe this further, we have now measured the relative fitness of a representative population of lon-knockout from short-term evolution in drug-free LB (population #3) that harbours GDA-2 against its ancestor (marked with DlacZ). These competition experiments were performed in LB (in which GDA-2 emerged spontaneously), as well as in LB supplemented with antibiotics at the concentrations used during the short term evolution.

Values of relative fitness, w (mean ± SD from 3 measurements), are provided below:

LB: 1.4 ± 0.2

LB + Trimethoprim: 1.6 ± 0.2

LB + Spectinomycin: 0.9 ± 0.2

LB + Erythromycin: 1.3 ± 0.3

LB + Nalidixic acid: 1.5 ± 0.2

LB + Rifampicin: 1.4 ± 0.2

These data show an increase in relative fitness in drug-free LB as would be expected. Interestingly, we also observe an increase in relative fitness in LB supplemented with antibiotics, except spectinomycin. This result supports the idea that GDA-2 is a “media adaptation” and provides a general fitness advantage to the lon knockout. However, as the reviewer pointed out, we should expect to see GDA-2 emerge spontaneously in antibiotic-supplemented media as well. We think that this does not happen as the fitness advantage of drug-specific mutations (GDAs or point mutations) far exceed the advantage of a media adaptation GDA. As a result, we only see the specific mutations that provide high benefit against the antibiotic at least over the relatively short duration of 20-25 generations. It is noteworthy the GDA-2 mutation does come up in LTMPR1 when it is passaged over >200 generations in drug-free media, but shows fluctuating frequency over time. We expect, therefore, that given enough time we may detect the GDA-2 mutations even in antibiotic-supplemented media.

We note, however, that a major caveat in the above fitness calculations is that we cannot be sure that the competing ancestor has no GDA-2 mutations during the course of the experiment. Thus, the above fitness values are only indicative and not definitive. We have therefore not included these data in the revised manuscript.

(2) It is unclear if the isolates WTMPR1 - 5 and LTMPR1 - 5 were pure clones. The authors write in L.488 "Colonies were randomly picked, cultured overnight in drug-free LB and frozen in 50% glycerol at -80C until further use." And in L. 492 "For long-term evolution, trimethoprim-resistant isolates LTMPR1, WTMPR4 and WTMPR5 were first revived from frozen stocks in drug-free LB overnight." From these descriptions, it is possible that the isolates contained a fraction of cells of other genotypes since colonies are often formed by more than one cell and thus, unless pure-streaked, a subpopulation is present and would in drug-free media be maintained. The possibility of pre-existing subpopulations is important for all statements relating to "reversal".

This is indeed a valid concern. As far as we can tell all our initial isolates (i.e. WTMPR1-5 and LTMPR1-5) are pure clones at least as far as SNPs are concerned. This is based on whole genome sequencing data that we have reported earlier in Patel and Matange, eLife (2021), where we described the evolution and isolation of WTMPR1-5 and the present study for LTMPR1-5. All SNPs detected were present at a frequency of 100%. For clones with GDAs, however, there is no way to eliminate a sub-population that has a lower or higher gene copy number than average from an isolate. This is because of the inherent instability of GDAs that will inevitably result in heterogeneous gene copy number during standard growth. In this sense, there is most certainly a possibility of a pre-existing subpopulation within each of the clones that may have reversed the GDA. Indeed, we believe that it is this inherent instability that contributes to their rapid loss during growth in drug-free media.

Minor Points:(1) L. 406. "allowing accumulation of IS transposases in *E. coli*" Please specify that it is the accumulation of transposase proteins (and not genes).

We have made this change.

(2) L. 221 typo. Known "to" stabilize.

We have made this change.

**Reviewer #2 (Recommendations for the authors):**
Most of my suggestions are found in the public review. I believe this to be a strong study, and some slight fixes can solidify its presence in the literature.

We have attempted to address the two main critiques by Reviewer 2. To simplify the understanding of our data, we have provided small schematics at various points in the paper to clarify the experimental pipelines used by us. We have also provided additional discussion situating our study in the emerging area of proteostasis and molecular evolution. We hope that our revisions have addressed these lacunae in our manuscript.

**Reviewer #3 (Recommendations for the authors):**
Major Points:(1) The manuscript is generally a bit difficult to follow. The writing is overly complicated and lacks clarity at times. It should be simplified and improved.

We have made several revisions to the text, as well as provided schematics in some of our figures which hopefully make our paper easier to understand.

(2) I cannot find the raw variant summary data for the lon strain evolution experiment in trimethoprim (after 25 generations). Were there any other mutations identified? If not, this should be explicitly stated in the text and the variant output summary from sequencing included as supplemental data.

We apologise for this oversight. We have now provided these data as Table 1.

(3) What is the trimethoprim IC50 of the starting (pre-evolution) strains (i.e. wt and lon)? I can't find this information, but it is critical to interpretation.

We had reported these values earlier in Matange N., J Bact (2020). Wild type and lon-knockout have similar MIC values for trimethoprim, though the lon mutant shows a higher IC50 value. We have now mentioned this in the results section (Line 100-101) and also provided the reference for these data.

(4) What was the average depth of coverage for WGS? This information is necessary to assess the quality of the variant calling, especially for the population WGS.

All genome sequencing data has a coverage at least 100x. We have added this detail to the methods section (Line 580-581).

(5) Five replicate evolution experiments (25 generations, or 7x 10% daily batch transfers) were performed in trimethoprim for the wt and lon strains. Duplication of the folA locus occurred in 1/5 and 4/5 experiments, respectively. It is not entirely clear what type of sampling was actually done to arrive at these numbers (this needs to be stated more clearly), but presumably 1 random colony was chosen at the end of the passaging protocol for each replicate. Based on this result, the authors conclude that folA duplication occurred more frequently in the lon strain, however, this is not rigorously supported by a statistical evaluation. With N=5, one cannot rigorously conclude that a 20% frequency and 80% frequency are significantly different. Furthermore, it's not entirely clear what the mechanism of resistance is for these strains. For example, in one colony sequenced (LTMPR5), it appears no known resistance mechanism (or mutations?) were identified, and yet the IC50 = 900 nM, which is also similar to other strains.

Indeed, we agree with the reviewer that we don’t have the statistical power to rigorously make this claim. However, since the lon-knockout showed us a greater frequency of GDA across 3 different environments we are fairly confident that loss of lon enhances the overall frequency for GDA mutations. This idea in also supported by a number of previous papers that related GDAs and IS-element transpositions with Lon, viz. Nicoloff et al, Antimicrob Agent Chemother (2007), Derbyshire et al. PNAS (1990), Derbyshire and Grindley, Mol Microbiol (1996). We have therefore not provided further justification in the revised manuscript.

We had indeed sampled a random isolate from each of the 5 populations and have added a schematic to figure 1 that provides greater clarity.

Having relooked at the sequencing data for LTMPR1-5 isolates (Table 1), we realised that both LTMPR4 and LTMPR5 harbour mutations in the pitA gene. We had missed this locus during the previous iteration of this manuscript and misidentified an mgrB mutations in LTMPR4. PitA codes for a metal-phosphate symporter. We have observed mutations in pitA in earlier evolution experiments with trimethoprim as well (Vinchhi and Yelpure et al. mBio 2023). Interestingly, in LTMPR5 there was a deletion of pitA, along with 17 other contiguous genes mediated by IS5. To test if loss of pitA is beneficial in trimethoprim, we tested the ability of a pitA knockout to grow on trimethoprim supplemented plates. Indeed, loss of pitA conferred a growth advantage to *E. coli* on trimethoprim, comparable to loss of mgrB, indicating that the mechanism of resistance of LTMPR5 may be due to loss of pitA. We have added these data to the Supplementary Figure 1 of the revised manuscript and provided a brief description in Lines 103-108. How pitA deficiency confers trimethoprim resistance is yet to be investigated. The mechanism is likely to be by activating some intrinsic resistance mechanism as loss of pitA also conferred a fitness benefit against other antibiotics. This work is currently underway in our lab and hence we do not provide any further mechanism in the present manuscript.

(6) Although measurement error/variance is reported, statistical tests were not performed for any of the experiments. This is critical to support the rigor and reproducibility of the conclusions.

We have added statistical testing wherever appropriate to the revised manuscript.

(7) Lines 150-155 and Figure 2E: Putting a wt copy of mgrB back into the WTMPR4 and LTMPR1 strains would be a better experiment to dissect out the role of mgrB versus the other gene duplications in these strains on fitness. Without this experiment, you cannot confidently attribute the fitness costs of these strains to the inactivation of mgrB alone.

We agree with the reviewer that our claim was based on a correlation alone. We have now added some new data to confirm our model (Figure 2 E, F). The costs of mgrB mutations come from hyperactivation of PhoQP. In earlier work we have shown that the costs (and benefit) of mgrB mutations can be abrogated in media supplemented with Mg^2+^, which turns off the PhoQ receptor (Vinchhi and Yelpure et al. mBio, 2023). We use this strategy to show that like the mgrB-knockout, the costs of WTMPR4, WTMPR5 and LTMPR1 can be almost completely alleviated by adding Mg^2+^ to growth media. These results confirm that the source of fitness cost of TMP-resistant bacteria was not linked to GDA mutations, but to hyperactivation of PhoQP.

(8) Figure 3F and G: Does the top symbol refer to the starting strain for the 'long-term' evolution? If so, why does WTMPR4 not have the mgrB mutation (it does in Figure 1)? Based on your prior findings, it seems odd that this strain would evolve an mgrB loss of function mutation in the absence of trimethoprim exposure.

We thank the reviewer for pointing this error out. We have made the correction in the revised manuscript.

(9) Figure 6A: If the marker is neutral, it should be maintained at 0.1% throughout the 'neutrality' experiment. In both plots, the proportion of some marked strains goes up and then down. This suggests either ongoing evolution (these competitions take place over 105 generations), or noisy data. I suspect these data are just inherently noisy. I don't see error bars in the plots. Were these experiments ever replicated? It seems that replicating the experiments might be able to separate out noise from signal and perhaps clarify this point and better confirm the hypothesis that the point mutants are more fit.

These experiments were indeed noisy and the apparent enrichment is most likely a measurement error rather than a real change in frequency of competing genotypes. We have now provided individual traces for each of the competing pairs with mean and SD from triplicate observations at each time point.

(10) Figure 6A: Please indicate which plotted line refers to which 'point mutant' using different colors. These mutants have different trimethoprim IC50s and doubling times, so it would be nice to be able to connect each mutant to its specific data plot.

We thank the reviewer for this suggestion. We have now colour coded the different strain combinations as suggested.

(11) Lines 284-285: I disagree that the IC50s are similar. The C-35T mutant has IC50 that is 2x that of LTMPR1. Perhaps more telling is that, compared to the folA duplication strain from the same time-point (which also carries the rpoS mutation), all of the point mutants have greater IC50s (~2x greater). 2-fold changes in IC50 are significant. It would seem that the point-mutants were likely not competing against LTMPR1 at the time they arose, so LTMPR1 might not be the best comparator if it was extinguished from the population early. I'm assuming this is why you chose a contemporary isolate (and, also, rpoS mutant) for the competition experiments. This should be explained more clearly.

We thank the reviewer for this comment. Indeed, the reviewer is correct about the rationale behind the use of a contemporary isolate and we have provided this clarification in the revised manuscript (Line 287-289). Also, the reviewer is correct in pointing out that a two-fold difference in IC50 cannot be ignored. However, the key point here would be in assessing the differences in growth rates at the antibiotic concentration used during competition (i.e. 300 ng/mL). We are unable to see a direct correlation between the growth rates and enrichment in culture indicating that the observed trends are unlikely to be driven by ‘level of resistance’ alone. We have added these clarifications to the modified manuscript (Lines 299-301)

Minor Points:(1) Line 13: Add a comma before 'Escherichia'

We have made this change.

(2) Line 14: Consider changing "mutations...were beneficial in trimethoprim" to "mutations...were beneficial under trimethoprim exposure"

We have made this change.

(3) Line 32: Is gene dosage really only "relative to the genome"? Is it not simply its relative copy number generally? Consider changing to "The dosage of a gene, or its relative copy number, can impact its level of expression..."

We have made this change.

(4) Line 38: The idea that GDAs are 1000x more frequent than point mutations seems an overgeneralization.

We agree with the reviewer and have softened our claim.

(5) Line 50: The term "hard-wired" is confusing. Please be more specific.

We have modified this statement to “…GDAs are less stable than point mutations….”.

(6) Line 52-53: What do you mean by "there is also evidence to suggest that...more common in bacteria than appreciated"? Are you implying the field is naïve to this fact? If there is "evidence" of this, then a reference should be included. However, it's not clear why this is important to state in the article. I would consider simply removing this sentence. Less is more in this case.

We have removed this statement.

(7) Lines 59-60: Enzymes catalyze reactions. Please also state the substrates for DHFR. Consider, "It catalyzes the NADPH-dependent reduction of dihydrofolate to tetrahydrofolate, and important co-factor for..."

We have made this change.

(8) Line 72: Please change to, "In *E. coli*, DHFR is encoded by folA." You do not need to state this is a gene, as it is implicit with lowercase italics.

We have made this change.

(9) Lines 72-86: This paragraph is a bit confusing to read, as it has several different ideas in it. Consider breaking it into two paragraphs at Line 80, "In this study,...". The first paragraph could just review the trimethoprim resistance mechanisms in *E. coli* and so would change the first sentence (Line 72) to reflect this topic: "In *E. coli*, DHFR is encoded by folA and several different resistance mechanisms have been characterized." Then, just describe each mechanism in turn. Also, by "hot spots" it would seem you are referring to "point mutations" in the gene that alter the protein sequence and cluster onto the 3D protein structure when mapped? Please be more specific with this sentence for clarity.

We have made these changes.

(10) Lines 92-93: Please also state the MIC value of the strain to specifically define "sub-MIC". Alternatively, you could also state the fraction MIC (e.g. 0.1 x MIC).

We have modified this statement to “…in 300 ng/mL of trimethoprim (corresponding to ~0.3 x MIC) for 25 generations.”

(11) Lines 95-96. Remove, "These sequencing have been reported earlier, ...(2021)". You just need to cite the reference.

We have made this change.

(12) Line 96: Remove the word "gene".

We have made this change.

(13) Figure 1 and Figure 4C: The color scheme is tough for those with the most common type of color blindness. Red/green color deficiency causes a lot of difficulty with Red/gray, red/green, green/gray. Consider changing.

We thank the reviewer for bringing this to our notice. We have modified the colour scheme throughout the manuscript.

(14) Figure 1: Was there a trimethoprim resistance mechanism identified for LTMPR5?

As stated by us in response to major comment #7, LTMPR5’s resistance seems to come from a novel mechanism involving loss of the pitA gene.

(15) Line 349-351: Please briefly define "lower proteolytic stability" as a relative susceptibility to proteolytic degradation and make sure it is clear to the reader that this causes less DHFR. This needs to be clarified because it is confusing how a mutation that causes DHFR proteolytic instability would lead to an increase in trimethoprim IC50. So, you also need to mention that some mutations can cause both increased trimethoprim inhibition and lower proteolytic stability simultaneously. It seems the Trp30Arg mutation is an example of this, as this mutation is associated with a net increase in trimethoprim resistance despite the competing effects of the mutation on enzyme inhibition and DHFR levels.

We thank the reviewer for this comment and agree that the text in the original manuscript did not fully convey the message. We have made modifications to this section (Lines 359-363) in the revised manuscript in agreement with the reviewer’s suggestions.